# AgentGym-RL: An Open-Source Framework to Train LLM Agents for Long-Horizon Decision Making via Multi-Turn RL

**Zhiheng Xi**[1,*,†]**, Jixuan Huang**[1,*]**, Chenyang Liao**[1,*]**, Baodai Huang**[1]**, Jiaqi Liu**[1]**,
Honglin Guo**[1]**, Yajie Yang**[1]**, Rui Zheng**[1]**, Junjie Ye**[1]**, Jiazheng Zhang**[1]**, Wenxiang Chen**[1]**,
Wei He**[1]**, Yiwen Ding**[1]**, Guanyu Li**[1]**, Zehui Chen**[2]**, Zhengyin Du**[2]**, Xuesong Yao**[2]**,
Yufei Xu**[2]**, Jiecao Chen**[2]**, Tao Gui**[1,3,4,†]**, Zuxuan Wu**[1,3,4]**, Qi Zhang**[1,3,4,†]**,
Xuanjing Huang**[1,3,4]**, Yu-Gang Jiang**[1,3,4]

[1]Fudan University [2]ByteDance Seed [3]Shanghai Key Laboratory of Multimodal Embodied AI
[4]Shanghai Collaborative Innovation Center of Intelligent Visual Computing
`zhxi22@m.fudan.edu.cn, {tgui, qz}@fudan.edu.cn`

## Abstract

Training LLM agents for complex multi-turn decision-making tasks requires extensive exploration within their environment, with reinforcement learning (RL) as a natural way. However, the open-source community currently lacks a unified RL framework capable of training agents from scratch across diverse and realistic environments. To bridge this gap, we introduce **AgentGym-RL**, a modular and decoupled framework specifically designed for RL-based agent in multi-turn decision-making tasks. It offers high flexibility and extensibility, supports mainstream RL algorithms, and spans a broad range of real-world scenarios. To effectively train agents for challenging tasks, we argue that they are required to expand external interactions with the environment, rather than relying solely on internal reasoning. Nevertheless, training agents for long-horizon interaction with vanilla methods often faces challenges like training instability. To this end, we propose **ScalingInter-RL**, a staged training approach for stable long-horizon RL training. It starts with short-horizon interaction to establish foundational policies and progressively expands them to encourage deeper exploration. Extensive experiments show that agents trained with our method achieve performance on par with—or even surpass—commercial counterparts like OpenAI o3 and Gemini-2.5-Pro across 27 tasks in diverse environments. We share key insights and release the full framework, including code and datasets, to empower the community in building the next generation of intelligent agents. Our framework is available at `https://github.com/WooooDyy/AgentGym-RL`.

## 1 Introduction

As Large Language Models (LLMs) rapidly advance (OpenAI, 2023; Anthropic, 2024; DeepSeek-AI et al., 2024; Team et al., 2023; Yang et al., 2025b), their applications have extended from chatbots to autonomous agents addressing long-horizon real-world decision-making tasks (Xi et al., 2025a; Moonshot AI, 2025). Analogous to human cognitive development, LLM agents are expected to acquire new knowledge and skills by actively exploring with the environment (Xi et al., 2025b; OpenAI, 2025).

Reinforcement learning (RL) is a natural choice for achieving this, demonstrating success in LLM reasoning (DeepSeek-AI et al., 2025; Jaech et al., 2024; Xi et al., 2024a; Trung et al., 2024; Team et al., 2025b; He et al., 2025). While recent efforts have sought to extend RL methodologies to develop LLM agents with multi-turn interaction capabilities (Zhou et al., 2024b; Chen et al., 2025;

---

*Equal contribution.
†Corresponding authors.

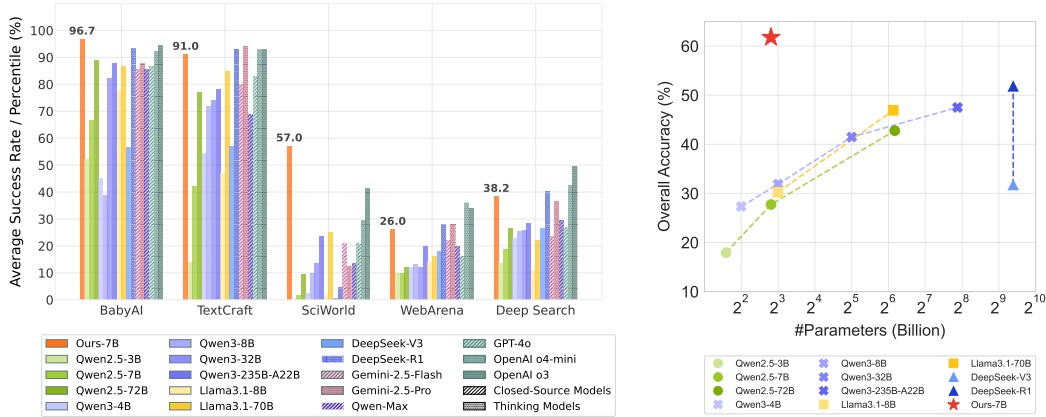

Figure 1: **Left**: Performance of proprietary models, open-source models, and our RL models across different agentic tasks. **Right**: Performance w.r.t model scale.

Wang et al., 2025; Qi et al., 2025; Jin et al., 2025b; Cao et al., 2025), they still struggle with limited task complexity and insufficient environmental diversity. Critically, the open-source community lacks unified RL framework capable of training agents from scratch across diverse, realistic environments.

To bridge this gap, we introduce **AgentGym-RL** (§3), a unified framework designed for training LLM agents through RL in multi-turn interactive decision-making tasks (Figure 2). With a modular and decoupled architecture, AgentGym-RL enables clean separation of agents, environments, and learning algorithms, offering high extensibility and flexibility for diverse research needs. The framework supports mainstream RL algorithms, and covers a wide range of real-world scenarios, e.g., web navigation (Zhou et al., 2024a; Yao et al., 2022), deep search (Wei et al., 2025; Jin et al., 2025b), digital games (Prasad et al., 2024; Fan et al., 2022), embodied tasks (Chevalier-Boisvert et al., 2019; Shridhar et al., 2021), and scientific tasks (Wang et al., 2022; Starace et al., 2025).

Furthermore, to enhance agents' ability to tackle challenging tasks, we argue that expanding their interactions with the environment is crucial, rather than relying solely on internal reasoning. However, our preliminary experiments show that directly training agents for long-horizon interaction often faces instability. To address this, we propose **ScalingInter-RL** (§4) based on AgentGym-RL. This progressively scaling interaction enables the agent to avoid repetitive and unproductive actions, enhance deeper exploration of environments, and ultimately achieve more effective and efficient task completion while maintaining training stability.

Extensive experiments (§5) demonstrate that ScalingInter-RL within AgentGym-RL framework delivers significant performance gains across 27 tasks spanning 5 diverse scenarios (Figure 1(Left)). Open-source models , e.g., Qwen-2.5-7B (Yang et al., 2024), achieve an average improvement of 33.65 points, matching or even surpassing larger commercial models such as OpenAI-o3 (OpenAI, 2025) and Gemini-2.5-Pro (Comanici et al., 2025). In addition, we conduct extensive analytical experiments to provide key insights (§6), showing that scaling both post-training and test-time interactions holds substantial potential for advancing agentic intelligence (Figure 1(Right)). We hope our work will serve as a valuable contribution to the community's progress.

## 2 PRELIMINARIES

### 2.1 FORMULATION

In this work, we study the multi-turn interactive decision-making tasks, i.e., agentic tasks, and we model them as a Partially Observable Markov Decision Process (POMDP) $(\mathcal{U}, \mathcal{S}, \mathcal{A}, \mathcal{O}, \mathcal{T}, r)$ like (Xi et al., 2025b; Zhou et al., 2024b), where $\mathcal{A}, \mathcal{U}, \mathcal{S}, \mathcal{O}, \mathcal{T} : \mathcal{S} \times \mathcal{A} \to \mathcal{S}, r \colon \mathcal{U} \times \mathcal{S} \to \mathbb{R}$ represents the instruction space, the state space, the action space, the observation space, the deterministic state transition function, and the reward function, respectively.

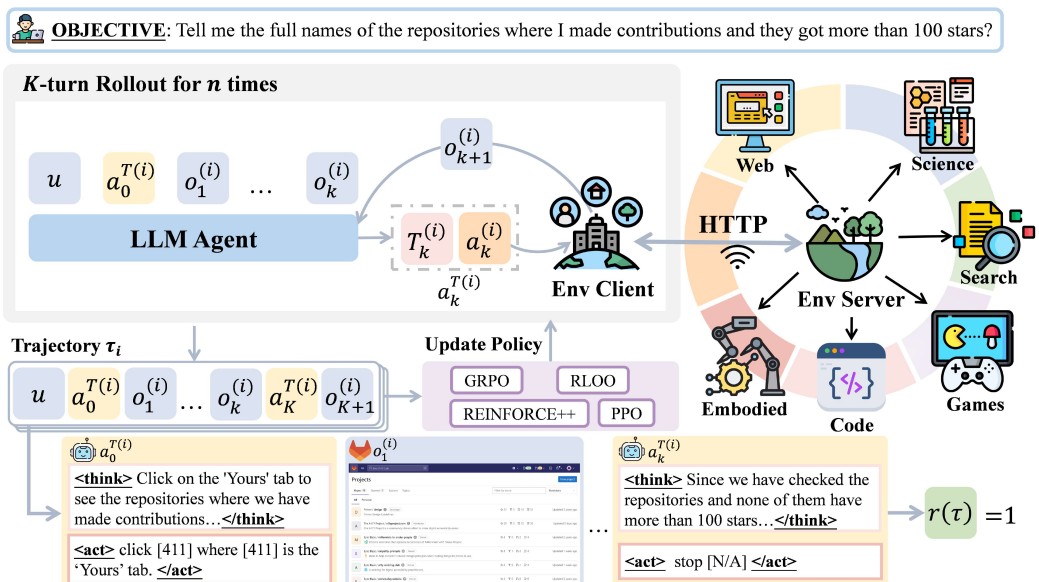

Figure 2: Overview of the AgentGym-RL framework. It features a decoupled, flexible, and extensible architecture, comprising three primary modules—the environment, the agent, and the training module. It supports diverse scenarios, environments, and algorithms.

Given a task instruction $u \in \mathcal{U}$, the agentic task requires the LLM agent to generate a sequence of actions $a_k^T \sim \pi_\theta(\cdot|s_k)$ based on its policy $\pi_\theta$ parameterized by $\theta$ to complete the given task, where $a_k \in \mathcal{A}$, and $s_k \in \mathcal{S}$, and $T$ is the reasoning path (Yao et al., 2023). The agent then receives an observation $o_k \in O$ from the environment, and the state is then transitioned to $\mathcal{T}(s_k, a_k) = s_{k+1}$. Finally after $N$ turns of interactions, the environment $e$ provides an outcome reward $r(\tau) \in [0, 1]$ to describe the completion of the multi-turn interactive decision-making tasks.

## 2.2 POLICY GRADIENT

We utilize policy gradient (PG) methods (Sutton et al., 1999) that optimizes our policy agent. They perform gradient ascent according to the objective $J(\theta)$, which is a function of the policy parameters $\theta$. Specifically, $J(\theta)$ represents the expected cumulative reward the agent anticipates receiving when following policy $\pi_\theta$ and interacting with the environment. Mathematically, this is expressed as the expectation of the total reward $r(\tau)$ over trajectories $\tau$ generated by the policy: $J(\theta) = \mathbb{E}_{\tau \sim \pi_\theta}[r(\tau)]$. To perform optimization on $J(\theta)$, we require the policy gradient $\nabla_\theta J(\theta)$. In the vanilla policy gradient methods, the policy gradient can be estimated by:

$$\nabla_\theta J(\theta) = \mathbb{E}_{\tau \sim \pi_\theta}\left[ r(\tau) \sum_{k=0}^{K} \nabla_\theta \log \pi_\theta(a_k|s_k) \right] \tag{1}$$

where $\pi_\theta$ is the policy parameterized by $\theta$, $\tau$ represents a trajectory consisting of a sequence of states and actions, $a_k$ and $s_k$ are the action and state at time step $k$, and $r(\tau)$ is the reward of the trajectory $\tau$. Mainstream RL algorithms for training LLMs include PPO (Schulman et al., 2017), GRPO (Shao et al., 2024), and REINFORCE++ (Hu, 2025)—all of which are integrated into our framework.

## 3 THE AGENTGYM-RL FRAMEWORK

### 3.1 ARCHITECTURE OVERVIEW

AgentGym-RL adopts a modular design with well-defined responsibilities for each module, allowing for extensibility. As shown in Figure 2, the framework is organized into three core modules.

**Environment module.** In this module, each environment is encapsulated as an independent service with the option of deploying multiple replicas to support parallel requests. An environment

```
# Stage 1: Generate responses
task_ids = expand(task_ids, sample_num)
envs = create_env_clients(task_ids, "webarena", base_url)

Do in parallel:
    for (env, task_id) in zip(envs, task_ids):
        env.reset(task_id)

handlers =[
    RolloutHandler().add_user_message(env.observe())
    for env in envs]

for i in range(max_rounds)
    prompts = [h.get_prompt() for h in handlers]
    responses = actor.generate(prompts)

    results = thread_safe_list()
    Do in parallel:
        for (env, response) in zip (envs, responses):
            results.append(env.step(response))

    for (h, r, res) in zip(handlers, responses, results):
        h.add_assistant_message(r)
        h.add_user_message(res.state)
        h.score = res.score

    if all_done(handlers): break
```

```
# Stage 2: Prepare experience
batch = gen_batch_from_rollout_handlers(handlers)
batch = actor.compute_log_prob(batch)
batch = reference.compute_ref_log_prob(batch)
batch = compute_advantages(batch, method="grpo")

# Stage 3: Actor training
actor.update_actor(batch)
```

Figure 3: Pseudocode demonstrating the example usage of our proposed framework (provided APIs marked orange), alongside a simplified diagram illustrating the agent-environment interaction and training pipeline.

client communicates with the environment server via HTTP and exposes APIs to the agent, including /observation to get the current observation, /available_actions to get the currently available actions, /step to perform an action, and /reset to reset the environment. Currently, AgentGym-RL covers five major scenario categories. This modular server–client design allows new environments to provide comprehensive environment and data support for LLM agent training.

**Agent module.** The agent module encapsulates the reasoning–action loop of LLM-based agents. It receives observations from the environment, performs reasoning over multiple turns, and outputs actions (e.g., invoking provided APIs). The module supports different prompting strategies and sampling configurations.

**Training module.** The training module provides a unified reinforcement learning (RL) pipeline that supports both online and offline algorithms, offering researchers a flexible foundation for large-scale LLM agent training. The module manages the entire RL lifecycle: trajectory collection, advantage estimation, policy optimization, and reward shaping.

**Workflow.** The overall workflow and pseudocode are shown in Figure 3. Given a batch of queries and initial environment states, the framework initializes multiple parallel environment clients. Each client serves a single agent, ensuring isolated execution. At every step, the agent generates an action, the environment returns the updated state and reward, and the trajectories are collected concurrently for training updates.

The entire training pipeline can be distributed across multiple nodes, leveraging both multi-process and multi-node parallelism. Efficient batching and asynchronous logging utilities ensure that system throughput scales with additional compute resources.

## 3.2 FEATURES AND CHARACTERISTICS

The AgentGym-RL framework is built on AgentGym (Xi et al., 2025b), which provides several basic interactive environments for LLM agents. We have further extended it in diversity of environments, algorithm support, engineering optimizations, open-source availability, and interaction visualization.

**Diverse scenarios and environments.** To build LLM agents capable of multi-turn decision-making, AgentGym-RL provides five heterogeneous environments spanning web navigation, deep search, digital games, embodied control, and scientific tasks. They exhibit significant variance in state space, action space, and reward structures. This cross-domain heterogeneity creates a testbed

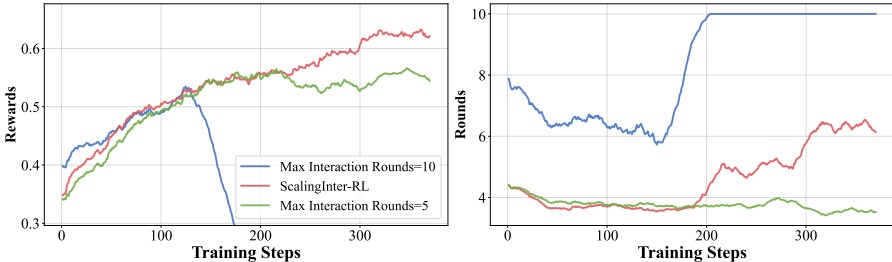

Figure 4: Training dynamics under different maximum interaction turns in Deep Search environment. Our ScalingInter-RL method progressively increases the interaction horizon, and ultimately achieves higher and more efficient long-term performance.

for training and evaluating research artifacts across diverse environments. A more detailed introduction of the environments we included is shown in Appendix C.

**Comprehensive algorithm support.** While the original AgentGym (Xi et al., 2025b) focused primarily on SFT, AgentGym-RL places online reinforcement learning at the core of its training stack. It allows agents to adapt through continual interaction with the environment and move beyond static demonstration corpora. The framework unifies mainstream RL algorithms such as PPO (Schulman et al., 2017), GRPO (Shao et al., 2024), RLOO (Chen et al., 2025) and REINFORCE++ (Hu, 2025) under a single interface, while also supporting complementary offline paradigms including SFT (Peng et al., 2023), DPO (Rafailov et al., 2023), and self-improvement (Xi et al., 2025b)).

**Engineering optimizations.** AgentGym-RL incorporates targeted engineering optimizations to support large-scale reinforcement learning research, with a focus on extensibility, scalability, and reliability. For extensibility, the framework adopts a modular plug-and-play design, allowing new environments to be integrated by simple inheritance from base classes. For scalability, we enhance both computational parallelism and long-horizon training efficiency by introducing optimizations like subprocess-based architecture and refined environment initialization routines. For reliability, we address critical issues such as memory leaks and flawed recursive implementations. A more detailed description of the engineering optimizations is shown in Appendix C.

**Open-source availability and Visualization support.** AgentGym-RL provides a unified framework with consistent evaluating metrics and reproducible training pipelines. It also offers turnkey scripts that automate the workflow from environment setup to final assessment, enabling reliable replication. Additionally, an interactive graphical UI (See Figure 10 in Appendix C) supports visualization of step-by-step inspection and replay of full trajectories.

## 4 SCALINGINTER-RL: SCALING INTERACTIONS FOR LLM AGENTS

**Motivation.** Inference–compute scaling in LLM reasoning shows that additional computation offers better performance (DeepSeek-AI et al., 2025; Jaech et al., 2024). However, given the interactive nature of agent tasks, we argue that **effective progress requires expanding external interactions with the environment, not merely internal reasoning**. To validate this, we investigate the impact of increasing the maximum number of interaction turns available to the agent, using several baseline

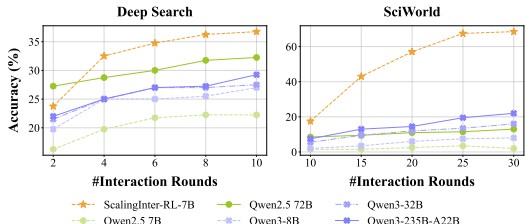

Figure 5: Scaling test-time interaction turns.

models on Deep Search and SciWorld environments. As shown in Figure 5, all models show improvement as the number of interaction turns increases, demonstrating that long-horizon interaction and sufficient exploration contribute to enhanced agentic performance. However, the performance gains of the baseline models plateau as the number of interactions continues to grow, indicating their limited capability to solve complex tasks through long-horizon interactions.

Figure 6: Training rewards in different environments leveraging AgentGym-RL framework with the ScalingInter-RL method.

To address this limitation, we further explore leveraging RL to enhance agents' capabilities in long-horizon scenarios. Specifically, we vary the maximum number of interaction turns during RL roll-outs and analyze the resulting training dynamics (Figure 4). We find that larger interaction horizons (e.g., 10 turns) enable deeper exploration but introduce training instability, often leading to training collapse, with the model exhibiting redundant interactions and unnecessary repetition. In contrast, shorter horizons provide stability but cap performance due to limited interaction turns. **Therefore, our core motivation is how to scale interactions at train-time in a stable and effective way.**

**Method.** To this end, we introduce **ScalingInter-RL** to stably optimize LLM agents for challenging tasks that require long-horizon interactions. The central idea of ScalingInter-RL lies in a progressive horizon-scaling strategy that gradually increases the number of interaction turns during RL training, as illustrated in Figure 9 (Appendix B).

Specifically, the objective is to maximize the expected final reward under a constrained interaction budget:

$$J(\theta) = \mathbb{E}_{\tau \sim \pi_\theta} \left[ r\left(\tau\right) \right],$$

where each trajectory $\tau = \left(a_0^T, o_1, a_1^T, \ldots, a_{K-1}^T, o_K\right)$ is sampled from the current policy $\pi_\theta$, with $K$ representing the total number of interaction turns, $T$ representing the reasoning path. To prevent the training collapse observed in the previously mentioned long-turn setting, we begin training with a short interaction horizon. By initially limiting the horizon, the agent focuses on exploitation, mastering fundamental task-solving skills through simpler tasks. This lays a solid foundation for stable training as the horizon gradually extends in later stages.

As training progresses, we introduce a monotonic schedule $\{h_1 < h_2 < \cdots < h_n\}$, where $h_t$ defines the maximum number of interaction turns allowed during phase t:

$$\tau_t \sim \pi_\theta\left(\tau \mid h_t\right), \quad \text{subject to } K_t \leq h_t.$$

The horizon $h_t$ is updated every $\Delta$ training steps according to a curriculum schedule:

$$h_{t+1} = h_t + \delta_h,$$

where $\delta_h$ is an adaptive increment. As the horizon expands, the agent is encouraged to explore the environment more deeply, thereby enhances the ability to efficiently acquire and leverage information through more interactions. This staged scaling approach allows the agent to make more intelligent decisions, enabling deeper exploration of the environment, and ultimately results in more effective task completion while ensuring training stability.

## 5 EXPERIMENTS

### 5.1 EXPERIMENTAL SETTINGS

**Scenarios, Environments and Tasks.** As mentioned before, we include five scenarios in AgentGym-RL. Specifically, we include WebArena (Zhou et al., 2024a) for web navigation, a RAG-based environment (Jin et al., 2025b; Joshi et al., 2017; Ho et al., 2020; Kwiatkowski et al., 2019; Mallen et al., 2022; Trivedi et al., 2022; Yang et al., 2018; Press et al., 2023) for deep search, TextCraft (Prasad et al., 2024) for digital games, BabyAI (Chevalier-Boisvert et al., 2019) for embodied tasks, and SciWorld (Wang et al., 2022) for scientific tasks.

Table 1: Evaluation results on Deep Search benchmark. For each group, the best result is in **bold**, and the second-best is underlined. SearchR1-it-v0.3 baseline uses Search-R1-v0.3 models (Jin et al., 2025a). See Appendix D for results of tasks on other scenarios.

| Model | NQ | TriviaQA | PopQA | HotpotQA | 2Wiki | Musique | Bamboogle | Overall |
|---|---|---|---|---|---|---|---|---|
| *Proprietary Models* | | | | | | | | |
| GPT-4o (Hurst et al., 2024) | 20.0 | **70.0** | 30.0 | 30.0 | 32.0 | 10.0 | 34.0 | 26.8 |
| Qwen-Max (Yang et al., 2024) | 24.0 | 52.0 | 26.0 | 24.0 | 16.0 | 17.0 | 36.0 | 29.5 |
| Gemini-2.5-Flash (Comanici et al., 2025) | 8.0 | 60.0 | 30.0 | 24.0 | 16.0 | 8.0 | 34.0 | 23.5 |
| OpenAI o4-mini (OpenAI, 2025) | 22.0 | 68.0 | 50.0 | 38.0 | 44.0 | 28.0 | 62.0 | 42.5 |
| OpenAI o3 (OpenAI, 2025) | **28.0** | **70.0** | **56.0** | **46.0** | **64.0** | **29.0** | **74.0** | **49.5** |
| Gemini-2.5-Pro (Comanici et al., 2025) | 22.0 | 62.0 | 38.0 | 28.0 | 48.0 | 19.0 | 56.0 | 36.5 |
| *Open-sourced Models ≥ 100B* | | | | | | | | |
| Qwen3-235B-A22B (Yang et al., 2025a) | 28.0 | 54.0 | 30.0 | 32.0 | 22.0 | 14.0 | 32.0 | 28.3 |
| DeepSeek-V3-0324 (DeepSeek-AI et al., 2024) | 28.0 | 60.0 | 24.0 | 28.0 | 18.0 | 11.0 | 34.0 | 26.5 |
| DeepSeek-R1-0528 (DeepSeek-AI et al., 2025) | **32.0** | **68.0** | **44.0** | **50.0** | **21.0** | **44.0** | **40.3** |
| *Open-sourced Models < 100B* | | | | | | | | |
| Qwen2.5-3B-Instruct (Yang et al., 2024) | 8.0 | 42.0 | 22.0 | 14.0 | 8.0 | 2.0 | 10.0 | 13.5 |
| Qwen2.5-7B-Instruct (Yang et al., 2024) | 18.0 | 54.0 | 20.0 | 18.0 | 6.0 | 4.0 | 26.0 | 18.8 |
| Qwen2.5-72B-Instruct (Yang et al., 2024) | 22.0 | 52.0 | 24.0 | 28.0 | 24.0 | 12.0 | **38.0** | 26.5 |
| Qwen3-4B (Yang et al., 2025a) | 18.0 | 58.0 | 26.0 | 24.0 | 26.0 | 5.0 | 20.0 | 22.8 |
| Qwen3-8B (Yang et al., 2025a) | 26.0 | 44.0 | 26.0 | 22.0 | 32.0 | 10.0 | 32.0 | 25.3 |
| Qwen3-32B (Yang et al., 2025a) | 24.0 | 54.0 | 22.0 | 36.0 | 28.0 | 11.0 | 20.0 | 25.8 |
| Llama-3.1-8B-Instruct (Dubey et al., 2024) | 16.0 | 26.0 | 12.0 | 6.0 | 2.0 | 4.0 | 18.0 | 11.0 |
| Llama-3.1-70B-Instruct (Dubey et al., 2024) | 20.0 | 44.0 | 22.0 | 22.0 | 18.0 | 9.0 | 32.0 | 22.0 |
| SearchR1-it-3B-v0.3_GRPO(Jin et al., 2025b) | 20.0 | 50.0 | 30.0 | 28.0 | 32.0 | 5.0 | 14.0 | 23.0 |
| SearchR1-it-7B-v0.3_GRPO(Jin et al., 2025b) | 24.0 | 52.0 | 30.0 | 22.0 | 34.0 | 6.0 | 26.0 | 25.0 |
| *Our RL Models* | | | | | | | | |
| AgentGym-RL-3B | 30.0 | 50.0 | 30.0 | 30.0 | **46.0** | 4.0 | 12.0 | 25.8 |
| AgentGym-RL-7B | 44.0 | 64.0 | 32.0 | 40.0 | 36.0 | **15.0** | 26.0 | 34.0 |
| ScalingInter-7B | **52.0** | **70.0** | **46.0** | **42.0** | 44.0 | 14.0 | 24.0 | **38.3** |

**Baselines and backbone models.** We leverage Qwen-2.5-3B and Qwen-2.5-7B (Yang et al., 2024) as our backbone models. Additionally, we introduce closed-source commercial models and strong open-source models as our baselines, as shown in Table 1. Both training and evaluation are conducted using ReAct (Yao et al., 2023) paradigm.

**Detailed settings of each environment.** Different environments have distinct observation spaces, action spaces, and reward structures. Due to space limitations, we provide detailed descriptions of the tools, APIs, and experimental settings for each environment in Appendix E.

## 5.2 MAIN RESULTS

The main results are shown in Figure 1, and the detailed results on Deep Search are shown in Table 1. See Appendix D for detailed results of tasks on other scenarios.

**Reinforcement learning generally improves agentic intelligence of open-source LLMs, bringing them on par with proprietary models.** As shown in Figure 1, our RL model outperforms other open-source models by a large margin. It also leads in average success rate over closed-source models like GPT-4o and Gemini-2.5-Pro across five different scenarios. This demonstrates the effectiveness of our framework in enabling models to learn and make decisions in complex tasks, narrowing the gap between open-source and proprietary models

**ScalingInter-RL significantly and consistently boosts performance.** We set phase transition points based on the total optimization steps in the RL process, rather than performing extensive hyperparameter tuning, as it has already proven effective. ScalingInter-RL consistently outperforms the baseline across various environments. For example, it improves WebArena performance by over 15 points, bringing it closer to closed-source commercial models. It also boosts TextCraft scores by nealy 50 points, achieving state-of-the-art results. These improvements show that our method effectively balances exploration and exploitation, enabling the model to interact more intelligently with the environment, adapt, and complete tasks.

**Post-training and test-time compute show higher scaling potential than model size.** As shown in Figure 1 (right), ScalingInter-RL with 7B parameters achieves an average success rate of 61.8%, significantly surpassing larger models like Llama3.1-70B (46.9%) and Qwen2.5-72B (42.8%). This shows that simply increasing model size provides limited performance gains, while increasing post-

training and inference-time compute offers better results, providing new insights for future scaling strategies.

**The environment plays a key role in the efficiency of reinforcement learning.** The effectiveness of AgentGym-RL depends on the environment and the type of feedback provided. In simulated worlds with clear rules and direct cause-and-effect relationships, such as TextCraft, BabyAI, and SciWorld, RL achieves the greatest performance improvements. For instance, SciWorld's score jumps from $1.50\%$ to $50.50\%$, a remarkable increase of almost 50 points. On the other hand, in more open-ended environments like WebArena and Deep Search, the performance gains from RL are more limited, due to the challenges of task complexity and potential noisy feedback. This provides valuable insights for the design of environmental feedback and reward structure in the future.

## 6 DISCUSSION

### 6.1 TEST-TIME SCALING FOR AGENTS

**Scaling interaction turns.** As shown in Figure 5, all models improve with more turns, showing that long-horizon interaction and sufficient exploration contribute to enhanced agentic performance. Moreover, the ScalingInter-RL-trained agent consistently surpasses the baseline by a substantial margin, further highlighting its ability in long-horizon scenaios and the effectiveness of our method.

**Scaling parallel sampling.** As shown in Figure 7, increasing the number of samples yields a marked improvement in Pass@K performance, signaling the downstream optimization potential of each model. The ScalingInter-RL trained model surpasses the baselines even with a small sampling budget, and as sampling increases, it continues to outperform the baseline in a stable and significant manner. Notably, in SciWorld, the ScalingInter-RL model's Pass@2 even surpasses all baselines' Pass@64, showcasing the compute-efficiency and superior optimization capability of our method.

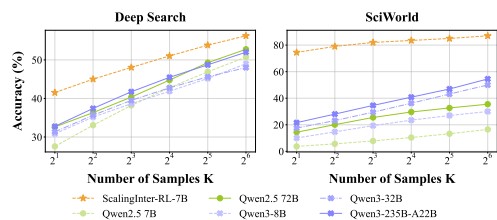

Figure 7: Pass@K performance.

### 6.2 PERFORMANCE OF DIFFERENT RL ALGORITHMS

We compare two mainstream RL algorithms for LLM post-training, i.e., GRPO and REINFORCE++. As shown in Table 2, GRPO consistently and substantially outperforms REINFORCE++ on the TextCraft, BabyAI, and Deep Search benchmarks. Notably, 3B-GRPO model even surpasses the 7B-REINFORCE++ model, highlighting an algorithmic advantage beyond model scale.

Table 2: Evaluation results of different RL algorithms.

| RL Algorithms | TextCraft | BabyAI | SearchQA |
|---|---|---|---|
| *Qwen2.5-3B-Instruct* | | | |
| GRPO | **75.00** | **93.33** | **25.75** |
| REINFORCE++ | 28.00 | 70.00 | 13.25 |
| *Qwen2.5-7B-Instruct* | | | |
| GRPO | **89.00** | **92.22** | **34.00** |
| REINFORCE++ | 73.00 | 84.44 | 24.00 |

The performance difference can be attributed to the way each algorithm calculates the advantage. GRPO calculates a baseline as the average value of multiple trajectories for a query, and then perform normalization, which helps reduce the impact of outliers from individual trajectories, leading to more robust optimization. In contrast, REINFORCE++ normalizes within a batch, which can lead to high-variance gradients.

### 6.3 ABLATION STUDY FOR SCALINGINTER-RL

We conduct detailed ablation studies on Deep Search task with respect to the initial number of interaction rounds, the stage transition frequency, and the interaction interval for ScalingInter-RL. The results are shown in Table 3. We find that ScalingInter-RL is not sensitive to these hyperparameters.

Table 3: Ablation study of ScalingInter-RL.

| Interact Turn List | Stage Transition Frequency | Performance |
|---|---|---|
| [5,8,10] | 100 | 38.3 |
| [5,8,10] | 75 | 37.8 |
| [5,8,10] | 125 | 38.5 |
| [3,8,13] | 100 | 36.8 |
| [8,10,12] | 100 | 37.6 |
| [5,10,15] | 100 | 39.1 |
| [5,7,9] | 100 | 37.8 |

## 6.4 EFFICIENCY ANALYSIS OF SCALINGINTER-RL

We analyze the efficiency of ScalingInter-RL by examining the training reward, the cumulative number of interaction rounds during training, and the total training time. As shown in Figure 8, we can observe that, thanks to the stage-based design, ScalingInter-RL is able to achieve relatively high rewards with comparatively high efficiency and reduced time and resource consumption.

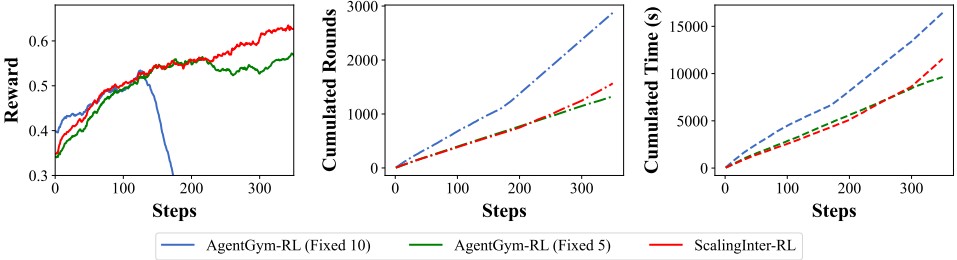

Figure 8: Analysis of computational resources and efficiency.

## 6.5 EXPERIMENTS OF APPLYING SCALINGINTER-RL TO MORE ALGORITHMS

We apply ScalingInter-RL to other algorithms and present the results in Table 4. We can observe performance differences across algorithms, which is consistent with previous work Hu (2025); Shao et al. (2024). In addition, it is clear that ScalingInter-RL brings performance improvements across different algorithms.

Table 4: Applying ScalingInter-RL to more algorithms.

| RL Algorithm | Method | TextCraft | BabyAI | SciWorld |
|---|---|---|---|---|
| Base Model | - | 42.00 | 66.67 | 1.50 |
| PPO | AgentGym-RL-7B | 68.00 | 86.66 | 10.83 |
|  | ScalingInter-7B | **71.00** | **90.00** | **25.69** |
| REINFORCE++ | AgentGym-RL-7B | 73.00 | 84.44 | 13.63 |
|  | ScalingInter-7B | **77.00** | **87.77** | **26.14** |

## 6.6 CASE STUDY

We provide a series of case studies on different tasks that highlight both the shortcomings of the base agent and the improvements achieved by our reinforcement learning agents in Appendix G.

**RL agent vs. Base agent.** RL-trained agents consistently outperform base agents by completing tasks more strategically. They can avoid unproductive loops and adapt to challenges. In the WebArena environment, Figures 15 and 16 show how RL optimization enhances web navigation. While base agents repeatedly click on ineffective interface elements without making progress, RL-trained agents recover from mistakes, escape deadlocks, and ultimately complete the task. In the BabyAI environment, Figures 12 and 13 illustrate a improvement in navigation capabilities. Unlike the base agent which exhibits repetitive movements, the RL agent demonstrates strategic backtracking, superior spatial reasoning, eventually accomplishes the task.

**Exception Cases.** To provide a balanced perspective, we also include two representative failure cases—in scientific reasoning and in efficient web navigation—that underscore areas for improvement. In the SciWorld environment, Figure 17 shows that while the RL agent can reach task-relevant states, it still struggles with execution. Two main issues are identified: substituting factual recall for necessary experimental procedures during debugging, and prematurely ending exploration by focusing solely on one animal. These failures demonstrate the agent's insufficient procedural understanding required for scientific analysis. In the WebArena environment, Figure 18 illustrate that

though the RL agent successfully reaching the correct target websites, it performs redundant actions such as unnecessary clicking, hovering and scrolling. These behaviors hinder effective information extraction, revealing a gap between state-reaching ability and precise, efficient action selection.

# 7 RELATED WORK

**Developing agents with large language models.** With the advancement of large language models (Achiam et al., 2023; Anthropic, 2024; Team et al., 2023), researchers have explored building agents for multi-turn decision-making (Xi et al., 2025a; Yao, 2024; Shen et al., 2026; Liu et al., 2026; Yang et al., 2026; Team et al., 2025a; Tang et al., 2025; Ye et al., 2025b;b). Current approaches mainly use prompting to invoke tools (Qin et al., 2025; Ye et al., 2025a;c), often enhanced with self-reflection (Shinn et al., 2023; Xi et al., 2024c; Xie et al., 2025; Renze & Guven, 2024; Yuan et al., 2025), long-horizon planning (Liu et al., 2023; Nayak et al., 2024; Prasad et al., 2024; Sun et al., 2023), and self-correction (Kamoi et al., 2024; Kumar et al., 2025). Multi-LLM workflows assign specialized roles to different models (Liang et al., 2024; Wu et al., 2023; Talebirad & Nadiri, 2023; Hong et al., 2024; Guo et al., 2025), but usually depend on proprietary models (e.g., OpenAI o3) and lack intrinsic agentic training. Another direction collects expert trajectories for imitation learning (Zhang et al., 2024; Zeng et al., 2024; Chen et al., 2023; 2024b), which grants skills like API use and planning but is costly, hard to scale, and limits self-improvement.

**Reinforcement learning for large language models.** Reinforcement learning is a crucial post-training technique for LLMs, supporting preference alignment (Ouyang et al., 2022; Zheng et al., 2023; Xia et al., 2024; Chen et al., 2024a; Ji et al., 2023), improved reasoning (Jaech et al., 2024; Trung et al., 2024; Xi et al., 2024a; DeepSeek-AI et al., 2025; Qwen Team, 2025; He et al., 2025; Xi et al., 2024b; 2025c), and new scaling strategies (DeepSeek-AI et al., 2025). Algorithms such as PPO (Schulman et al., 2017), GRPO (Shao et al., 2024), REINFORCE++ (Hu, 2025), and RLOO (Chen et al., 2025) have been widely adopted. Yet most efforts like DeepSeek-R1 focus on single-turn tasks, limiting multi-turn interaction with complex environments. Recent advances extend RL to self-reflection (Xie et al., 2025), tool use (Ye et al., 2024), and long-horizon interaction (Zhou et al., 2024b; Chen et al., 2025; Wang et al., 2025; Qi et al., 2025; Jin et al., 2025b; Cao et al., 2025), but face challenges in scalability, task diversity, and optimization stability. To address this, we present a unified RL framework for multi-turn decision-making across diverse environments, and introduce ScalingInter-RL, an interaction-scaling method that stabilizes training and enhances agent performance.

**Scaling inference compute for language models.** Increasing inference compute both at test time and during RL rollouts yields strong scaling effects (Jaech et al., 2024; DeepSeek-AI et al., 2025; xAI, 2025; Snell et al., 2024; Xi et al., 2025d). Techniques like long-chain-of-thought reasoning (Snell et al., 2024; Xi et al., 2024c), majority voting (Li et al., 2024; Wang et al., 2023), best-of-N sampling (Chow et al., 2025; Jinnai et al., 2024), beam search (Xie et al., 2023; Zhu et al., 2024), and Monte Carlo tree search (Chi et al., 2024; Gan et al., 2025). Zhu et al. (2025) address inference-scaling for LLM agents but they do not investigate inference scaling in RL. TTI (Shen et al., 2025) teaches compute allocation via rejection sampling. By contrast, we use on-policy RL (e.g., GRPO, REINFORCE++) to scale interactions without restricting compute to "thinking" or "acting", letting the agent adaptively allocate extra compute to improve exploration, skill acquisition, and performance.

# 8 CONCLUSION

In this work, we present AgentGym-RL, a unified reinforcement learning framework for training LLM agents in long-horizon, multi-turn decision-making tasks. The framework offers diverse environments and scenarios, integrates mainstream RL algorithms, and provides a high degree of extensibility, making it a versatile and powerful resource for the community. Building on this, we introduce ScalingInter-RL, a staged training approach that progressively scales agent–environment interactions and achieves strong final performance. Extensive experiments demonstrate the effectiveness of both the framework and the method. We hope our work offers valuable insights and supports the development of next-generation intelligent agents.

## ETHICS STATEMENT

This paper presents AgentGym-RL, a unified framework that enable stable reinforcement learning of LLM agents for diverse real-world multi-turn tasks. It further proposes ScalingInter-RL, a training approach designed for exploration-exploitation balance and stable RL optimization. We firmly state that this work is intended for ethical and constructive purpose. While no immediate societal harms are identified, proactive measures should ensure responsible deployment to mitigate potential misuse or unintended consequences.

## REPRODUCIBILITY STATEMENT

We claim our detailed experiment setting in Appendix E. In addition, we upload anonymized versions of our data and code in a Zip file with a Readme file to ensure easy reproduction of all reported results.

## ACKNOWLEDGEMENT

The authors wish to thank the anonymous reviewers for their helpful comments. This work was partially funded by the Science and Technology Commission of Shanghai Municipality (No.24511103100), National Natural Science Foundation of China (No.62476061, 62576106, 62376061), Shanghai Rising-Star Program (23QA1400200), and Natural Science Foundation of Shanghai (23ZR1403500).

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

## A  The Use of Large Language Models

LLMs are utilized in this manuscript for partial grammatical checks and language polishing. The authors are fully responsible for the final content.

## B  Illustration of ScalingInter-RL

Our ScalingInter-RL is illustrated in Figure 9.

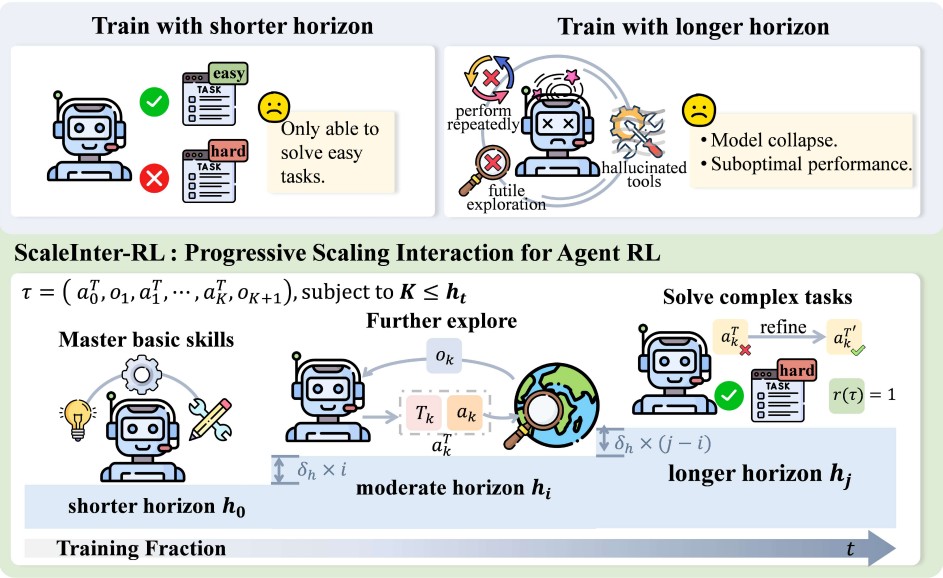

Figure 9: Illustration of ScalingInter-RL to scale up agent-environment interactions progressively.

## C  Details of the Features and Characteristics of AgentGym-RL Framework

**Diverse scenarios and environments.**  The environment's anisotropic complexity ensures that successful policies must develop domain-agnostic reasoning capabilities rather than task-specific heuristics, making it an ideal benchmark for evaluating the generalization robustness of our ScalingInter-RL methodology. It includes:

- **Web Navigation**: Interacting with dynamic websites for tasks such as booking flights or extracting structured information, which requires agents to follow instructions, interpret textual and visual content, manipulate dynamic interfaces, and plan multi-step actions.
- **Deep Search**: Performing multi-step, goal-directed queries with tools like browsers or Python interpreters, demanding strong information-seeking, multi-hop reasoning, long-term memory, and knowledge synthesis across sources.
- **Digital Games**: Exploring and solving problems in interactive game-like environments, emphasizing real-time decision-making, strategy development, and adaptability to complex, dynamic settings.
- **Embodied Tasks**: Controlling virtual or physical bodies for navigation, manipulation, and task execution, which calls for goal-directed planning, spatial reasoning, and robust perception–action grounding.
- **Scientific Tasks**: Conducting experiments and solving problems in physically grounded, knowledge-intensive settings, requiring precise execution, dynamic interpretation of feedback, evidence-based reasoning, and iterative hypothesis refinement.

**Extensibility** is essential for advancing research, enabling a framework to incorporate new environments, agent architectures, and training methods without modifying existing components. AgentGym-RL adopts a modular and decoupled design, where the core components, Environment, Agent, and Training are fully plug-and-play. This extensible design allows researchers to incorporate novel environments through simple inheritance from base classes (e.g., `BaseEnvClient`), and implementing the required methods such as `reset()`, `step()`, and `observe()`.

**Scalability** addresses the growing demands of large-scale reinforcement learning training that requires massive data processing and extended interaction sequences. AgentGym-RL implements comprehensive architectural optimizations to enhance both computational parallelism and training duration capabilities. For example, we replaced WebArena's single-browser-per-process design with a subprocess-based architecture enabling concurrent Chromium instance management. These optimizations collectively enable effective scaling for large-scale training and diverse experimental requirements.

**Reliability** ensures consistent operation during extended multi-turn agent training by preventing failures and managing critical resources effectively. AgentGym-RL implements targeted optimizations to address system vulnerabilities that could disrupt long-horizon training. For example, we resolved TextCraft's memory leak in its recursive `crafting_tree` implementation, where redundant self-replication caused exponential memory growth and training crashes by refactoring the recursion to eliminate redundant copies. These optimizations provide a stable foundation for uninterrupted operation across extended interaction sequences.

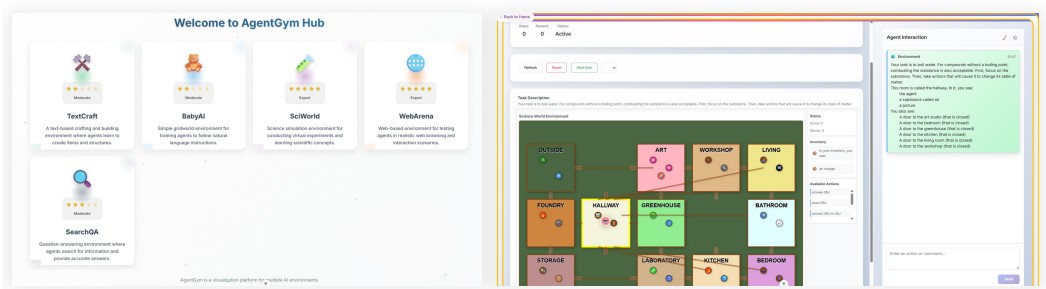

Figure 10: An overview of the visualized user interface of our framework.

**Standardized evaluation and reproducibility.** AgentGym-RL is designed to be user-friendly for the community. To systematically address reproducibility challenges in LLM-based reinforcement learning, AgentGym-RL institutes a standardized evaluation process and reproducible training pipelines. This design enforces uniform metrics and consistent experimental procedures to ensure fair comparisons. We provide easy-to-setup reproduction scripts that automate the entire workflow, from environment configuration to final evaluation. This design enables researchers to replicate prior findings with high fidelity and significantly lowers the barrier for building upon existing work, thereby promoting verifiable research standards.

**Visualized observability and analysis.** An interactive graphical UI supports step-by-step inspection and replay of full interaction trajectories, visualizing observations, internal reasoning, and actions to reveal performance and failure modes and accelerate iterative development.

## D    DETAILED TASK PERFORMANCE ACROSS ENVIRONMENTS

**Web navigation.** As shown in Table 5, our models demonstrate highly competitive performance on the WebArena benchmark. In particular, the ScalingInter-7B model achieves an overall accuracy of 26.00%, significantly surpassing top-tier proprietary models like GPT-4o (16.00%) and performing on par with larger models like DeepSeek-R1-0528 (28.00%) and Gemini-2.5-Pro (28.00%). Furthermore, another 7B model of ours, AgentGym-RL-7B, also achieved an overall score of 22.00%, surpassing the performance of GPT-4o. This strong overall performance is underpinned by ScalingInter-7B's state-of-the-art proficiency in structured web navigation, where it achieved scores

Table 5: Evaluation results on WebArena benchmark. For each group, the best result is in **bold**, and the second-best is underlined. In the first row, G & R means GitLab and Reddit.

| Model | Shopping | CMS | Maps | G & R | Overall |
|---|---|---|---|---|---|
| *Proprietary Models* | | | | | |
| GPT-4o | 20.00 | 13.33 | 10.00 | 20.00 | 16.00 |
| Qwen-Max | 20.00 | 13.33 | 20.00 | 30.00 | 20.00 |
| Gemini-2.5-Flash | 26.67 | 20.00 | 10.00 | 30.00 | 22.00 |
| OpenAI o4-mini | **33.33** | **26.67** | 20.00 | 70.00 | **36.00** |
| OpenAI o3 | **33.33** | 0.00 | **40.00** | **80.00** | 34.00 |
| Gemini-2.5-Pro | 26.67 | **26.67** | 0.00 | 60.00 | 28.00 |
| *Open-sourced Models $\geq$ 100B* | | | | | |
| Qwen3-235B-A22B | 20.00 | **20.00** | 20.00 | 20.00 | 20.00 |
| DeepSeek-V3-0324 | 20.00 | 13.33 | 10.00 | 30.00 | 18.00 |
| DeepSeek-R1-0528 | **33.33** | 6.67 | **30.00** | **50.00** | **28.00** |
| *Open-sourced Models < 100B* | | | | | |
| Qwen2.5-3B-Instruct | 13.33 | 6.67 | 10.00 | 10.00 | 10.00 |
| Qwen2.5-7B-Instruct | 14.29 | 6.67 | 0.00 | 16.67 | 9.76 |
| Qwen2.5-72B-Instruct | 13.33 | 13.33 | 0.00 | 20.00 | 12.00 |
| Qwen3-4B | 13.33 | 6.67 | 10.00 | 20.00 | 12.00 |
| Qwen3-8B | 20.00 | 20.00 | 0.00 | 10.00 | 14.00 |
| Qwen3-32B | 20.00 | 6.67 | **20.00** | 0.00 | 12.00 |
| Llama-3.1-8B-Instruct | 13.33 | 0.00 | **20.00** | **30.00** | 14.00 |
| Llama-3.1-70B-Instruct | 26.67 | 6.67 | **20.00** | 10.00 | 16.00 |
| *Our RL Models* | | | | | |
| AgentGym-RL-3B | 20.00 | 26.67 | 10.00 | 10.00 | 18.00 |
| AgentGym-RL-7B | 20.00 | **33.33** | 0.00 | **30.00** | 22.00 |
| ScalingInter-7B | **33.33** | 26.67 | **20.00** | 20.00 | **26.00** |

of 33.33% in Shopping and 26.67% in CMS, matching the best performance among all models in these categories. However, a significant performance gap remains when compared to the top-performing OpenAI o3 (34.00%) and o4-mini (36.00%), a disparity almost entirely concentrated in the "GitLab & Reddit" sub-task.

**Deep search.** The evaluation results in Table 1 show the importance of sophisticated reasoning abilities, where proprietary models—particularly the OpenAI 'o' series—currently set the performance benchmark, with OpenAI o3 achieving the highest overall score of 49.50%. Against this competitive landscape, our models demonstrate exceptional performance. Specifically, our ScalingInter-7B model achieved an excellent overall score of 38.25%, not only surpassing top-tier proprietary models like GPT-4o (26.75%) and Gemini-2.5-Pro (36.50%) but also performing comparably to the strongest open-source model, DeepSeek-R1-0528 (40.25%). Its strengths are particularly salient in key domains: it achieved the highest score overall on the NQ task (52.00%) and tied for first place on TriviaQA (70.00%) with GPT-4o. Furthermore, our AgentGym-RL-7B (34.00%) and AgentGym-RL-3B (25.75%) models also delivered strong results, each significantly outperforming open-source counterparts of similar or even larger scales. These results provide strong evidence that our reinforcement learning approach effectively unlocks the model's inherent reasoning capabilities, enabling it to reach or even exceed the performance of elite reasoning models in key scenarios–crucially, without the need for explicit additional long-reasoning.

**Digital game.** The TextCraft benchmark effectively assesses model capabilities across a wide spectrum of difficulty, as detailed in Table 6. At shallow depths (Depth 1), tasks are largely solved by top models. Conversely, the challenge becomes nearly insurmountable at maximum complexity (Depth 4), creating a performance cliff for most agents. It is at these intermediate and highest difficulties that the efficacy of our models becomes particularly evident. Our ScalingInter-7B model achieves an outstanding overall score of 91.00%, puuting it on par with the top-tier proprietary and large open-source models (93.00%-94.00%). Critically, it is one of only a few models to achieve a non-zero score at Depth 4, scoring 33.33% and demonstrating a unique robustness at maximum complexity. Our AgentGym-RL-7B also excels with a score of 89.00, surpassing prominent models

Table 6: Evaluation results on TextCraft benchmark. For each group, the best result is in **bold**, and the second-best is underlined.

| Model | Depth 1 | Depth 2 | Depth 3 | Depth 4 | Overall |
|---|---|---|---|---|---|
| *Proprietary Models* | | | | | |
| GPT-4o | **100.00** | 87.80 | 64.00 | 0.00 | 83.00 |
| Qwen-Max | 93.55 | 75.61 | 36.00 | 0.00 | 69.00 |
| Gemini-2.5-Flash | **100.00** | 95.12 | 40.00 | 0.00 | 80.00 |
| OpenAI o4-mini | **100.00** | **100.00** | **84.00** | 0.00 | 93.00 |
| OpenAI o3 | **100.00** | **100.00** | **84.00** | 0.00 | 93.00 |
| Gemini-2.5-Pro | **100.00** | **100.00** | **84.00** | 33.33 | **94.00** |
| *Open-sourced Models $\geq$ 100B* | | | | | |
| Qwen3-235B-A22B | **100.00** | **100.00** | **84.00** | 0.00 | **93.00** |
| DeepSeek-V3-0324 | 80.65 | 53.66 | 40.00 | 0.00 | 57.00 |
| DeepSeek-R1-0528 | **100.00** | **100.00** | **84.00** | 0.00 | **93.00** |
| *Open-sourced Models < 100B* | | | | | |
| Qwen2.5-3B-Instruct | 35.48 | 7.32 | 0.00 | 0.00 | 14.00 |
| Qwen2.5-7B-Instruct | 80.65 | 41.46 | 0.00 | 0.00 | 42.00 |
| Qwen2.5-72B-Instruct | 96.77 | 85.37 | 48.00 | 0.00 | 77.00 |
| Qwen3-4B | 87.10 | 36.59 | 12.00 | 0.00 | 45.00 |
| Qwen3-8B | **100.00** | 78.05 | 40.00 | **33.33** | 74.00 |
| Qwen3-32B | 90.32 | 92.68 | 72.00 | **33.33** | 85.00 |
| Llama-3.1-8B-Instruct | 74.19 | 56.10 | 4.00 | 0.00 | 47.00 |
| Llama-3.1-70B-Instruct | **100.00** | **100.00** | **84.00** | 0.00 | **93.00** |
| *Our RL Models* | | | | | |
| AgentGym-RL-3B | **100.00** | 90.24 | 28.00 | 0.00 | 75.00 |
| AgentGym-RL-7B | **100.00** | 97.56 | 72.00 | 0.00 | 89.00 |
| ScalingInter-7B | **100.00** | 97.56 | 76.00 | **33.33** | 91.00 |

like GPT-4o (83.00%). The benefit of our RL training is especially dramatic for smaller models, where AgentGym-RL-3B obtains a score of 75.00%, vastly outperforming similarly-sized models like Qwen2.5-3B-Instruct (14.00%). These results showcase that our RL approach elevates our models to achieve competitive performance on complex, sequential decision-making tasks.

**Embodied tasks.** As demonstrated in Table 7, our RL model achieves state-of-the-art (SOTA) performance on the BabyAI benchmark, with an overall score of 96.67%, which is competitive with the leading proprietary models such as o3 and o4-mini. Notably, our ScalingInter-7B model attains the highest overall accuracy of 96.67%, outperforming top-tier models such as OpenAI o3 (94.44%) and GPT-4o (86.67%). This exceptional performance is driven by ScalingInter-7B's consistent mastery of diverse sub-tasks, achieving perfect scores of 100% in GoTo, ActionObjDoor (AOD), and SynthLoc, and strong results of 80% in both FindObjS7 (Find) and OneRoomS20 (Room). Similarly, our AgentGym-RL-7B and AgentGym-RL-3B models demonstrate robust capabilities, reaching overall accuracies of 92.22% and 93.33%, respectively, and securing perfect scores in GoTo and AOD tasks. Compared to other open-sourced models, such as Qwen3-235B-A22B (87.78%) and DeepSeek-R1-0528 (93.33%), our RL-based models maintain consistently high performance while effectively handling more challenging sub-tasks like Room and Find, where many LLMs exhibit notable variability. Overall, these results highlight the strength of our RL-based approaches, particularly ScalingInter-7B, in achieving state-of-the-art performance on both structured navigation and object-interaction tasks in the BabyAI benchmark.

**Scientific Scenario.** Our experiments on the SciWorld benchmark, summarized in Table 8, demonstrate the advanced performance of our RL-trained models. Our ScalingInter-7B model establishes a new state-of-the-art with an overall score of 57.00%, which significantly surpasses all open-source and proprietary models, including the next-best proprietary model, OpenAI o3 (41.50%). This superior performance is primarily attributed to high scores in the "Find" (88.64%) and "Test-Cond" (55.42%) sub-tasks. Furthermore, our AgentGym-RL-7B model also shows strong capabilities, securing the second-highest overall score (50.50%) and achieving the top score in "Test-Cond" (59.04%). These results highlight the effectiveness of our RL method for training agents in exploration and procedural execution tasks. However, our findings also identify a critical limitation shared across all evaluated models. The "Chem-Mix" sub-task proved to be intractable, with every model,

Table 7: Evaluation results on BabyAI benchmark. For each group, the best result is in **bold**, and the second-best is underlined. In the first row, AOD means ActionObjDoor, Find means FindObjS7, Room means OneRoomS20, SLoc means SynthLoc.

| Model | GoTo | Pickup | AOD | Find | Room | SLoc | Overall |
|---|---|---|---|---|---|---|---|
| *Proprietary Models* | | | | | | | |
| GPT-4o | 92.73 | 80.00 | **100.00** | **80.00** | **60.00** | 60.00 | 86.67 |
| Qwen-Max | 92.73 | 80.00 | 80.00 | 60.00 | **60.00** | 80.00 | 85.56 |
| Gemini-2.5-Flash | 92.73 | 86.67 | 80.00 | 20.00 | **60.00** | 100.00 | 85.56 |
| OpenAI o4-mini | 96.36 | **100.00** | **100.00** | **80.00** | 40.00 | 80.00 | 92.22 |
| OpenAI o3 | **98.18** | 93.33 | 100.00 | **80.00** | **60.00** | 100.00 | **94.44** |
| Gemini-2.5-Pro | 94.55 | 93.33 | **100.00** | 40.00 | **60.00** | 60.00 | 87.77 |
| *Open-sourced Models* | | | | | | | |
| Qwen3-235B-A22B | 89.09 | **86.67** | **100.00** | **80.00** | 60.00 | **100.00** | 87.78 |
| DeepSeek-V3-0324 | 67.27 | 53.33 | 0.00 | 20.00 | 40.00 | 60.00 | 56.67 |
| DeepSeek-R1-0528 | **98.18** | **86.67** | **100.00** | 60.00 | **80.00** | **100.00** | **93.33** |
| *Open-sourced Models* | | | | | | | |
| Qwen2.5-3B-Instruct | 61.82 | 40.00 | 20.00 | 60.00 | 40.00 | 20.00 | 52.22 |
| Qwen2.5-7B-Instruct | 70.91 | 66.67 | 60.00 | **80.00** | 60.00 | 20.00 | 66.67 |
| Qwen2.5-72B-Instruct | 92.73 | 93.33 | **100.00** | 60.00 | 60.00 | 80.00 | 88.89 |
| Qwen3-4B | 60.00 | 60.00 | 40.00 | 40.00 | 40.00 | 20.00 | 54.44 |
| Qwen3-8B | 43.64 | 20.00 | 40.00 | 40.00 | 40.00 | 40.00 | 38.89 |
| Qwen3-32B | 87.27 | 80.00 | **100.00** | 60.00 | 40.00 | 80.00 | 82.22 |
| Llama-3.1-8B-Instruct | 85.45 | 60.00 | **100.00** | **80.00** | 60.00 | 40.00 | 77.78 |
| Llama-3.1-70B-Instruct | 89.09 | 86.67 | **100.00** | 60.00 | 60.00 | **100.00** | 86.67 |
| *Our RL Models* | | | | | | | |
| AgentGym-RL-3B | **100.00** | **100.00** | **100.00** | 60.00 | 60.00 | 60.00 | 93.33 |
| AgentGym-RL-7B | **100.00** | 93.33 | **100.00** | 60.00 | 60.00 | 60.00 | 92.22 |
| ScalingInter-7B | **100.00** | 93.33 | **100.00** | **80.00** | **80.00** | **100.00** | **96.67** |

including our top performers, scoring zero. This uniform result indicates a systemic challenge for current language models in tasks requiring complex scientific reasoning and multi-step chemical simulation, marking this as a crucial area for future research.

# E    IMPLEMENTATION DETAILS AND SETTINGS OF EACH ENVIRONMENT

We conduct all the experiments on NVIDIA A100 GPUs and Ascend 910B NPUs. The remaining part of this section shows detailed setting of different environments.

## E.1    WEB NAVIGATION SCENARIO

**Tools and APIs.**    In web navigation scenario, the agent simulates human interaction with web pages to ultimately complete the task. WebArena(Zhou et al., 2024a) supports these interactioins through a set of tool APIs, allowing agents to perform a variety of real-world tasks, including online shopping, engaging in discussions on Reddit, collaborating on software development via GitLab, and managing store content through a CMS. In addition to these online platforms, WebArena also provides three utility-style tools: a map for navigation and location-based information search, a calculator, and a scratchpad for note-taking.

A query case of web navigation is shown below:

> **Web Navigation Example**
>
> You are an autonomous intelligent agent tasked with navigating a web browser. You will be given web-based tasks. These tasks will be accomplished through the use of specific actions you can issue.

Table 8: Evaluation results on SciWorld benchmark. For each group, the best result is in **bold**, and the second-best is underlined. In the first row, Test-Cond. means test-conductivity, Chem-Mix means chemistry-mix.

| Model | Measure | Test-Cond. | Find | Chem-Mix | Lifespan | Overall |
|---|---|---|---|---|---|---|
| *Proprietary Models* | | | | | | |
| GPT-4o | 15.09 | 6.02 | 38.64 | 20.00 | 73.33 | 21.00 |
| Qwen-Max | 9.43 | 0.00 | 34.09 | 20.00 | 40.00 | 13.50 |
| Gemini-2.5-Flash | 11.32 | 0.00 | 54.55 | 0.00 | 80.00 | 21.00 |
| OpenAI o4-mini | 20.75 | 14.46 | 47.73 | 0.00 | **100.00** | 29.50 |
| OpenAI o3 | **47.17** | **25.30** | **56.82** | **40.00** | 66.67 | **41.50** |
| Gemini-2.5-Pro | 9.43 | 0.00 | 29.55 | 0.00 | 46.67 | 12.50 |
| *Open-sourced Models $\geq$ 100B* | | | | | | |
| Qwen3-235B-A22B | **11.32** | **4.82** | **59.09** | **20.00** | **66.67** | **23.50** |
| DeepSeek-V3-0324 | 0.00 | 0.00 | 2.27 | 0.00 | 0.00 | 0.50 |
| DeepSeek-R1-0528 | 1.89 | 0.00 | 11.36 | 0.00 | 20.00 | 4.50 |
| *Open-sourced Models < 100B* | | | | | | |
| Qwen2.5-3B-Instruct | 3.77 | 0.00 | 0.00 | 0.00 | 0.00 | 1.00 |
| Qwen2.5-7B-Instruct | 1.89 | 0.00 | 0.00 | 0.00 | 13.33 | 1.50 |
| Qwen2.5-72B-Instruct | 7.55 | 1.20 | 15.91 | 20.00 | 40.00 | 9.50 |
| Qwen3-4B | 0.00 | 0.00 | 0.00 | 0.00 | 33.33 | 2.50 |
| Qwen3-8B | 9.43 | 0.00 | 18.18 | 0.00 | 46.67 | 10.00 |
| Qwen3-32B | 5.66 | 1.20 | 31.82 | 0.00 | 66.67 | 14.00 |
| Llama-3.1-8B-Instruct | 9.43 | 0.00 | 4.55 | 20.00 | 0.00 | 4.00 |
| Llama-3.1-70B-Instruct | 24.53 | 4.82 | 40.91 | **40.00** | 86.67 | 25.00 |
| *Our RL Models* | | | | | | |
| AgentGym-RL-3B | 20.75 | 28.92 | 0.00 | 0.00 | 66.67 | 22.50 |
| AgentGym-RL-7B | 24.53 | **59.04** | 65.91 | 0.00 | 66.67 | 50.50 |
| ScalingInter-7B | **33.96** | 55.42 | **88.64** | 0.00 | 73.33 | **57.00** |

**Available Information:**

- **User's objective**: The task to complete
- **Accessibility tree**: Simplified webpage representation, providing key information.
- **Current URL**: The active page's address
- **Open tabs**: Currently available tabs
- **Previous action**: Last performed action

**Action Categories:**
*Page Operations:*

- `click [id]`: Click element with ID
- `type [id] [content] [0|1]`: Input text (1=press Enter)
- `hover [id]`: Hover over element
- `press [key_comb]`: Simulate key press (e.g., Ctrl+v)
- `scroll [down|up]`: Scroll page direction

*Tab Management:*

- `new_tab`: Open new tab
- `tab_focus [tab_index]`: Switch to tab
- `close_tab`: Close current tab

*URL Navigation:*

- `goto [url]`: Navigate to URL
- `go_back`: Return to previous page

> - `go_forward`: Advance to next page
>
> *Completion:*
>
> - `stop [answer]`: Submit final answer (or "N/A" if you believe the task is impossible to complete)
>
> Homepage: If you want to visit other websites, check out the homepage at `http://homepage.com`.
>
> ---
>
> Objective: Among the top 10 post in "books" forum, show me the book names from posts that recommand a single book.

**Settings.** We include five subtasks: E-commence, Reddit, Gitlab, OpenStreetMap (Map), and online store content management system (CMS), comprising a total of 372 training queries and 50 testing queries. These are selected from the origin WebArena dataset, which contains 812 queries across three categories: Information Seeking, Site Navigation, and Content & Config. To facilitate efficient parallel rollout, we exclude the Content & Config tasks, which involve insert, update and delete operations that change the state of the websites. We set the maximum number of agent-environment interactions to 15 turns in both AgentGym-RL training and evaluation. In ScalingInter-RL, we gradually increase the maximum number of interactions transition from 8 to 12 and then to 15, with each transition occurring every 80 step. We employ GRPO as the main RL algorithm with a learning rate of $5 \times 10^{-7}$ and a KL coefficient of $1 \times 10^{-3}$. For each query, we sample 4 distinct trajectories using a temperature of 1.0.

### E.2 DEEP SEARCH SCENARIO

**Tools and APIs.** The deep search senario features a search engine–based environment equipped with specialized tools and APIs supporting the interaction with search engines. These APIs enable agents to dynamically generate search queries during the reasoning process, retrieve relevant information from external sources, and incorporate the retrieved information into subsequent reasoning steps. This setting allows agents to engage in complex reasoning processes that involve iterative searching and information integration, thereby enhancing their capability to solve intricate problems where external knowledge is essential.

A query case of Deep Search is shown below:

> **Deep Search Example**
>
> You must always reason inside <think>...</think> first; if you lack knowledge, issue a <search>...</search> and then stop; do not generate <information> or <answer> yet; wait for external input between <information>...</information> before continuing; resume only when new <information> is given; do not skip steps or anticipate answers early.
>
> ---
>
> Question: Who got the first Nobel Prize in Physics?

**Settings.** We include queries from 7 datasets following the setup of Search-R1 (Jin et al., 2025b): NQ (Kwiatkowski et al., 2019), TriviaQA (Joshi et al., 2017), PopQA (Mallen et al., 2022), HotpotQA (Yang et al., 2018), 2wiki (Ho et al., 2020), Musique (Trivedi et al., 2022), and Bamboogle (Press et al., 2023). To ensure fair comparison and balanced evaluation, we randomly sample 400 examples from the development sets of NQ, TriviaQA, PopQA, HotpotQA, 2wiki, Musique, and Bamboogle. The maximum number of agent-environment interactions is set to 10 turns in evaluation, and to 5 turns in AgentGym-RL training. In ScalingInter-RL, the maximum number of interactions is initially set to 5, increased to 8 at step 200, and further to 10 at step 300 We employ GPRO as the main algorithm for reinforcement learning setups with a learning rate of $1 \times 10^{-6}$, a KL coefficient of $1 \times 10^{-3}$, and a sampling temperature of 1.0. We sample 8 distinct trajectories for a single query.

### E.3 DIGITAL GAMES SCENARIO

**Environments, Tools and APIs.** As for digital games, we introduce TextCraft(Prasad et al., 2024), a text-based game environment mirroring Minecraft. The APIs in TextCraft include crafting, inventory management, and dynamic narrative generation. These APIs allow agents to execute predefined crafting recipes, manipulate inventory contents, navigate virtual spaces, dynamically generate quests and sub-tasks based on natural language objectives, and recursively decompose complex tasks into achievable sub-goals.

A query case of TextCraft can be seen below:

---

**TextCraft Example**

You are given few useful crafting recipes to craft items in Minecraft. Crafting commands are of the format "craft [target object] using [input ingredients]".
Every round I will give you an observation, you have to respond an action based on the state and instruction. You can "get" an object (ingredients) from the inventory or the environment, look-up the game inventory by "inventory", or "craft" (target) using any of the crafting commands. You can use ONLY these crafting commands provided, do not use your own crafting commands. However, if the crafting command uses a generic ingredient like "planks", you can use special types of the same ingredient e.g. "dark oak planks" in the command instead.

- - - - - - - - - - - - - - - - - - - - - - - - - - - - - - - - - - - - - - - - - - - - - - - - - - - -

Goal: Craft flint and steel.

---

**Settings.** In TextCraft, task difficulty is measured by the maximum depth of the corresponding crafting tree. In practice, the benchmark contains tasks with crafting trees of depths 1, 2, 3, and 4. Accordingly, we divide the entire task set into four subsets based on these depths. We set the maximum number of interactions to 20 turns in evaluation, and set to 30 turns in AgentGym-RL training. In ScalingInter-RL, we gradually increase the maximum number of interactions transition from 10 to 20 and then to 30, with each transition occurring every 100 step. We employ GRPO as the main RL algorithm with a learning rate of $1 \times 10^{-6}$, a KL coefficient of $1 \times 10^{-3}$, and a sampling temperature of 1.0. We sample 8 distinct trajectories for a single query.

### E.4 EMBODIED SCENARIO

**Tools and APIs.** We introduce the BabyAI environment as a representative setting for embodied tasks. It provides APIs that allow agents to navigate a controllable grid world using natural language instructions. Through these APIs, agents can perform actions such as moving objects, unlocking doors, and interacting with the environment in response to textual commands.

A query case of BabyAI can be seen below:

---

**BabyAI Example**

You are an exploration master that wants to finish every goal you are given. Every round I will give you an observation, and you have to respond an action and your thought based on the observation to finish the given task. You are placed in a room and you need to accomplish the given goal with actions.
You can use the following actions:
- turn right - turn left - move forward - go to *obj id* - pick up *obj id*
- go through *door id*: *door* must be an open door.
- toggle and go through *door id*: *door* can be a closed door or a locked door. If you want to open a locked door, you need to carry a key that is of the same color as the locked door.
- toggle: there is a closed or locked door right in front of you and you can toggle it.

- - - - - - - - - - - - - - - - - - - - - - - - - - - - - - - - - - - - - - - - - - - - - - - - - - - -

Your goal: Go to the red ball.

---

**Settings.** Following the original implementation, we divide the tasks into six subsets based on the final goal. We set the maximum number of interactions to 20 turns in both evaluation and AgentGym-RL training. In ScalingInter-RL, we gradually increase the maximum number of interactions transition from 6 to 13 and then to 20, with each transition occurring every 100 step. We employ GRPO as the main RL algorithm with a learning rate of $1 \times 10^{-6}$, a KL coefficient of $1 \times 10^{-3}$, and a sampling temperature of 1.0. We sample 8 distinct trajectories for a single query.

### E.5 SCIENTIFIC SCENARIO

**Tools and APIs.** SciWorld(Wang et al., 2022) is an agent environment for scientific tasks. It provides APIs that are designed to support scientific exploration through text-driven reasoning cycles. These APIs empower agents to conduct experiments by interacting with various scientific apparatus and performing actions like measuring temperature, connecting electrical circuits, and mixing chemicals.

A query case of SciWorld can be seen below:

---

**SciWorld Example**

You are an agent for science world. Every round I will give you an observation, you have to respond an action based on the observation to finish the given task.

- - - - - - - - - - - - - - - - - - - - - - - - - - - - - - - - - - - - - - - - - - - - - - - - - - - - -

Your task is to boil water. For compounds without a boiling point, combusting the substance is also acceptable. First, focus on the substance. Then, take actions that will cause it to change its state of matter.

---

**Settings.** We select 8 subsets of tasks from the original SciWorld environment. We set the maximum number of agent-environment interactions to 20 turns in both AgentGym-RL training and evaluation. In ScalingInter-RL, we gradually increase the maximum number of interactions transition from 10 to 15 and then to 20, with each transition occurring every 200 step. We employ GRPO as the main RL algorithm with a learning rate of $1 \times 10^{-6}$, a KL coefficient of $1 \times 10^{-3}$, and a sampling temperature of 1.0. We sample 8 distinct trajectories for a single query.

## F ANALYSIS ABOUT THE INSTABILITY OF LONG-HORIZON TRAINING

From a theoretical perspective, for a given query, an LLM agent performs multiple rounds of interaction, where each round is a ReAct step (Reasoning with an Action). Each step consists of multiple tokens, so the entire interaction produces a large volume of tokens, which makes the task very challenging for an agent that lacks sufficient environmental context and basic capabilities.

We conduct an empirical analysis, as shown in the Figure 11. We find that, in terms of training reward, long-horizon training tends to collapse, mainly due to the uncertainty introduced by long-horizon interactions and operations. At the same time, we observe that this phenomenon is often accompanied by abnormal gradients, where the grad norm exhibits multiple spikes, which empirically confirms its instability.

## G TRAJECTORY EXAMPLES AND VISUALIZATIONS OF OUR RL AGENT

This appendix provides additional trajectory visualizations and detailed analysis across multiple environments. The figures illustrate the behaviors of both baseline and RL-trained agents, highlighting the RL model's superior performance in exploration, task execution, and interaction patterns, while also revealing common failure modes that remain.

**Enhanced navigation.** Figure 12 demonstrates a notable improvement in navigation capabilities within BabyAI environment. While the base agent exhibited suboptimal behavior characterized by repetitive movement patterns-going through previously explored locations without developing a strong search strategy for completion, the RL agent manifested more effective exploration strategy.

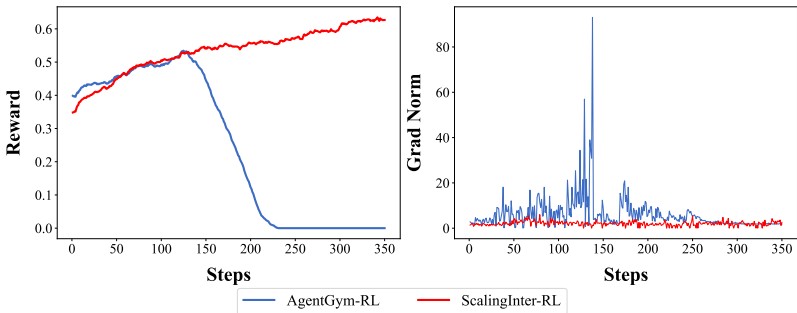

Figure 11: Empirical analysis on the instability of long-horizon training.

It demonstrated strategic backtracking capabilities, systematically exiting through doorways before selecting alternative pathways, ultimately accessing a green door that provided direct access to the target blue box. This highlights the RL agent's superior ability in spatial reasoning and its ability to circumvent unproductive behavioral loops.

**Compositional Task Mastery.** Figure 14 exemplifies the successful application of reinforcement learning to complex scientific task execution. The base agent exhibited fundamental deficiencies in task interpretation, misusing non-interactive objects and generating invalid actions. In contrast, the RL-optimized agent demonstrated comprehensive task understanding through its systematic approach: correctly identifying and manipulating a living thing (the banana tree), executing appropriate inventory management operations, navigating multi-room environments with obstacle resolution capabilities and successfully completing the objective by depositing the tree in the designated purple box. This highlights the RL agent's enhanced capabilities in reasoning, planning, and sequential task execution within compositional problem spaces.

**Adaptive Web Navigation Strategies.** Figure 15 and Figure16 illustrates the emergence of web navigation capabilities through reinforcement learning optimization. The base agent persistently interacted with non-responsive interface elements, specifically engaging in repetitive clicking behaviors on ineffective targets without recognizing the futility of these actions. Our RL-trained agent exhibited markedly superior adaptive behavior: it successfully implemented error recovery mechanisms when encountering a "Page not found" error, subsequently utilizing the search box to locate the "pittsburgh" forum, identifying contextually relevant content within trending posts, and completing the subscription task successfully—demonstrating enhanced robustness in error handling, purposeful navigation strategies, and the ability to maintain task focus while avoiding unproductive behavioral patterns.

**Limitations in Scientific Scenario.** Figure 17 reveals fundamental procedural execution failures that persist in SciWorld task completion despite the RL agent's ability to reach task-relevant game states. These instances exemplify two distinct failure modalities: first, when confronted with interaction failures requiring systematic debugging, the agent inappropriately substitutes direct factual recall for the intended experimental procedure; second, the agent demonstrates insufficient systematic exploration, as evidenced by its premature task termination after navigating to the outdoor environment and focusing only on the chameleon egg rather than analyzing all available animals that the task demands. These failures collectively indicate that the model lacks the deep procedural understanding necessary for executing rigorous scientific comparative analyses.

**Over-Interaction Patterns in Web Navigation.** Figure 18 demonstrates a prevalent failure mode of excessive and inefficient interaction sequences during web navigation tasks. Despite successfully navigating to the correct target websites in both illustrated cases, the RL agent engages in superfluous interaction patterns—including redundant clicking, unnecessary hovering, and excessive scrolling—that impede successful information extraction from the target pages. These behavioral patterns suggest that the reinforcement learning process failed to instill the precision and efficiency required for optimal task completion, indicating a gap between state-reaching capabilities and effective action selection within those states.

---

### BabyAI

**Task:** go to the blue box

**Obs 0:** In front of you in this room, you can see several objects: There is a yellow closed door 1 2 steps in front of you and 1 steps to your right. There is a green closed door 1 3 steps in front of you and 3 steps to your left. The room has walls around you. You are facing a wall 3 steps away. You are not carrying anything.

**Available actions:** ["turn left", "turn right", "move forward", "toggle and go through yellow closed door 1", "toggle and go through green closed door 1", "go to yellow closed door 1", "go to green closed door 1", "check available actions"]

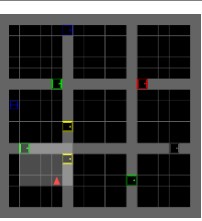

---

### Base Model

**Thought 1:** To reach the blue box, I need to move forward, but there is a yellow closed door in front of me, blocking the way. …it's more efficient to go through the yellow closed door first.
**Act 1:** toggle and go through yellow closed door 1
**Obs 1:** There is a green closed door 1 6 steps in front of you and 2 steps to your right. The room has walls around you. You are facing a wall 6 steps away.

**Thought 2:** … Now that I have gone through the yellow closed door, Since I don't have any keys, I need to toggle and go through this door to find the blue box.
**Act 2:** toggle and go through green closed door 1
**Obs 2:** There is a grey closed door 1 4 steps in front of you and 3 steps to your left. The room has walls around you. You are facing a wall 6 steps away.
**……**
**Thought 4:** …my only option is to toggle it and attempt to go through.
**Act 4:** toggle and go through red closed door 1
**Obs 4:** The room has walls around you. You are facing a wall 6 steps away.
**Act 5:** move forward
**Act 6:** go through red open door 1
**Act 7:** go through grey open door 1
**……**
**Act 13:** go through grey open door 1
**Act 14:** go through red open door 1
**……**
**Thought 15:** Given that the room has walls around me and I am not carrying anything, it appears I have reached the end of this path. I have already explored this area thoroughly and there are no more objects or doors to interact with.
**Act 15:** check available actions
**……**
**Thought 20:** Turning left again has brought me to a wall, confirming that I have explored all possible directions in this room.
**Act 20:** check available action

**Score: 0.0**

---

### Our RL Model

**Thought 1:** … find the blue box
**Act 1:** toggle and go through yellow closed door 1
**Obs 1:** There is a green closed door 1 6 steps in front of you and 2 steps to your right. The room has walls around you. You are facing a wall 6 steps away.

**Thought 2:** … find the blue box
**Act 2:** go to green closed door 1
**Obs 2:** There is a green closed door 1 right in front of you 1 steps away. The room has walls around you.

**Thought 3:** … find the blue box
**Act 3:** toggle and go through green closed door 1
**Obs 3:** There is a grey closed door 1 4 steps in front of you and 3 steps to your left. The room has walls around you. You are facing a wall 6 steps away.
**……**
**Act 4:** go to grey closed door 1
**Act 5:** toggle and go through grey closed door 1
**Act 6:** go to red closed door 1
**Act 7:** turn right
**Act 8:** turn right
**Act 9:** go through grey open door 1
**Act 10:** turn right
**Act 11:** go through green open door 1
**Act 12:** go through yellow open door 1
**……**
**Act 13:** toggle and go through green closed door 1.
**Obs 13:** There is a blue box 1 4 steps in front of you and 1 steps to your left. There is a green closed door 1 6 steps in front of you and 3 steps to your right. The room has walls around you. You are facing a wall 6 steps away.
**Act 14:** go to blue box 1.

**Score: 1.0**

---

Figure 12: Comparison of our RL agent with the base agent on the BabyAI task. Our RL model significantly outperforms the base model, successfully navigating to the blue box while the base model fails to complete the task.

## H    COMPARATIVE CASE ANALYSIS

Figure 19 shows a case from the TextCraft environment. The short-horizon agent fails because it uses incorrect quantities (violating crafting constraints) and drifts from the goal of crafting an orange bed. In contrast, our agent succeeds by adhering to constraints and managing resources strategically: it proactively checks inventory, adaptively explores alternatives (e.g., crafting intermediate items), and efficiently acquires materials to resolve shortages. This showcases stronger long-term planning and underscores the effectiveness of our method.

Figure 20 shows a case from SearchQA environment. The short-horizon agent fails because it only performs superficial searches. When encountering misleading information (such as the references to the Jefferson family), it does not conduct further verification or cross-check the family lineage. ScalingInter-RL agent, on the other hand, actively adjusts its strategy after making an error. Through

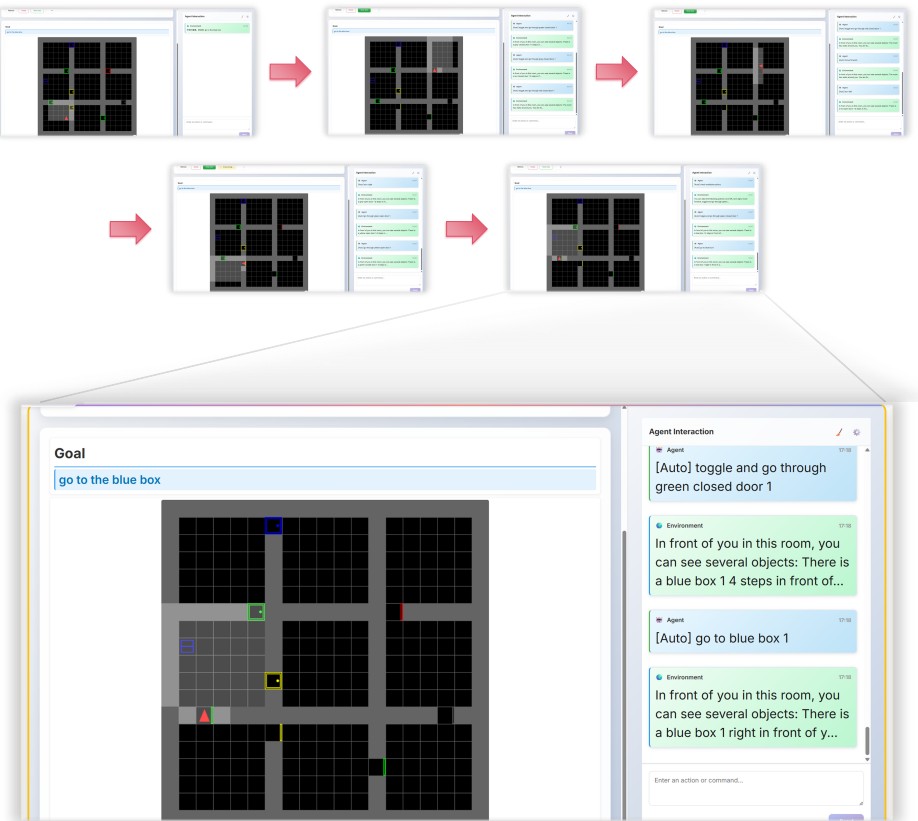

Figure 13: Trajectory visualization in the Baby AI task, highlighting the agent's path through the environment, action execution, and feedback.

multiple turns of search, it verifies hypotheses, filters out irrelevant information, and further traces the parentage of Sally Hemings. It gradually narrows the search scope step by step, and ultimately identifies the key genealogical fact (that Sally Hemings's father is John Wayles), allowing it to arrive at the correct answer.

Figure 21 shows a case from TextCraft environment. The long-horizon agent failed because it did not follow the task objective and directly craft mossy stone bricks. Instead, it attempted an action less relevant to the goal—crafting mossy cobblestones—which actually moved it farther away from completing the target task. In contrast, our agent maintained goal-relevant exploration, correctly found the path to craft mossy stone bricks, and thus completed the task more efficiently. This case suggests that the long-horizon agent may suffer from over-exploration, engaging in redundant exploration in the environment while neglecting goal-directed progress.

Figure 22 shows a case from TextCraft environment. The short-horizon agent failed because it did not explore the environment appropriately and sufficiently. Instead, it performed excessive internal reasoning and incorrectly assumed it needed to obtain iron ingots by mining iron ore, even though this was not required by the instructions at the current difficulty level. As a result, it overlooked a more direct way to obtain the ingredients needed to craft iron chestplates. In contrast, our agent explored the environment correctly and sufficiently, recognized that it could directly obtain the required ingredients from the environment, and—based on its exploration results—collected iron nuggets, then crafted iron ingots, and completed the task. This case indicates that the short-horizon agent may face an excessive reasoning issue: it relies too heavily on internal reasoning, makes incorrect assumptions about the environment state, and fails to interact with the environment to explore.

**SciWorld**

**Task:** Determine if sodium chloride is electrically conductive. The sodium chloride is located around the workshop. First, focus on the sodium chloride. If it is electrically conductive, place it in the yellow box. If it is electrically nonconductive, place it in the purple box.

**Obs 0:** Your task is to This room is called the hallway. In it, you see: the agent, a substance called air, a picture. You also see: A door to the art studio (that is closed), A door to the bedroom (that is closed), A door to the greenhouse (that is closed), A door to the kitchen (that is closed), A door to the living room (that is closed), A door to the workshop (that is closed)

**Base Model**

**Thought 1:** The task is to determine if sodium chloride is electrically conductive. The sodium chloride is located in the workshop, so I need to go there first. I will use the "focus on" Act to signal my intent and then proceed to the workshop.
**Act 1:** focus on workshop
**Obs 1:** You focus on the workshop.

**Thought 2:** I have focused on the workshop, so the sodium chloride should now be ready. I will try to focus on the sodium chloride next.
**Act 2:** focus on sodium chloride
**Obs 2:** No sodium chloride found.

**Thought 3:** Sodium chloride is not in the workshop yet. I'll try looking around again to find it.
**Act 3:** look around
**Obs 3:** You are in the hallway. …

**Score: 0.0**

**Our RL Model**

**Thought 1:** My task is to determine if sodium chloride is conductive and place it in a yellow box if conductive or in a purple box if nonconductive. Let's find some sodium chloride around the workshop.
**Act 1:** look around
**Obs 1:** …
**……**
**Thought 4:** The door to the workshop is now open. Let's go through to the workshop.
**Act 4:** go to the workshop
**Obs 4:** You move to the workshop.

**Thought 5:** I have gone to the workshop. Let's see if there's any sodium chloride around here or around the area.
**Act 5:** look around
**Obs 5:** This room is called the workshop. In it, you see: … a substance called sodium chloride…

**Thought 6:** I found some sodium chloride around the workshop. Let's pick it up and focus on it to determine if it's conductive.
**Act 6:** pick up the sodium chloride
**Obs 6:** move the sodium chloride to the inventory.
**Thought 7:** I picked up the sodium chloride and added it to my inventory. Let's focus on the sodium chloride.
**Act 7:** focus on the sodium chloride
**Obs 7:** You focus on the sodium chloride.

**Thought 8:** I focused on the sodium chloride. Sodium chloride is an ionic compound and is electrically nonconductive. Let's put it in the purple box.
**Act 8:** put the sodium chloride in the purple box
**Obs 8:** move the sodium chloride to the purple box.

**Score: 100**

Figure 14: Performance comparison of the RL agent versus the base model on the SciWorld task. RL agent achieves a perfect score by successfully moving a banana to a purple box, while the base model scores0.0.

## WebArena

**Task:** Open the thread of a trending post on the forum "pittsburgh" and subscribe.
**Obs 0: URL: http://reddit.com/**    Tab 0 (current): Postmill

            [1] RootWebArea 'Postmill' focused: True
                 [232]  link 'Forums'
                 [233] link 'Wiki'
                 [241] searchbox 'Search query'
                 [242] link 'Notifications (0)'
                 [243] link 'Submit'

### Base Model

**Thought 1: ...** We need to navigate to the "pittsburgh" forum first. To do this, we can click on the "Forums" link in the header...
**Act 1:** click [232]
**Obs 1: ...** [1958] searchbox 'Search query'...

**Thought 2:** ...we can see that "AskReddit" and "relationship_advice" are listed, but "pittsburgh" is not. We need to search for the "pittsburgh" forum using the search box.
**Act 2:** type [1958] [pittsburgh] [1]
**Obs 2: ...** [2951] link 'From Florida to Pittsburgh : Is $100,000 home budget in Pittsburgh on disability income feasibl'...

**Thought 3: ...** Since the current page is not clearly indicating a trending post, we should navigate back to the list of forums and find the "pittsburgh" forum to check for trending posts.
**Act 3:** goto [http://reddit.com/r/pittsburgh]
**Obs 3:** ... [4739] StaticText 'Utilities have gone up significantly everywhere over the past 6 months with rising energy costs (both natural gas and electric), so I wouldn't necessarily link that to Pittsburgh.'...
**......**
**Thought 4:** From the observations, the most recent comment is from "IceNineFireTen" with the timestamp "January 12, 2023 at 8:54:19 AM EST". This is a relatively recent post, and it might be a trending post. We should open this thread to verify.
**Act 4:** click [4739]
**Obs 4:** Nothing changed.
**......**
**Act 5-15:** click [4739]
**Obs 5-15:** Nothing changed.

**Score: 0.0**

### Our RL Model

**Thought 1:** ...We need to navigate to the "pittsburgh" forum first.
**Act 1:** click [66]
**Obs 1:** "Page not found"

**Thought 2:** ...the current page is a "Page not found" error page. We need
to navigate back to the homepage or**....**
**Act 2:** go_back
**Obs 2:** ...

**Thought 3:** ... there is a link to the "Forums" section which we can use to navigate to the "pittsburgh" forum.
**Act 3:** click [65]
**Obs 3:** ... [500] searchbox 'Search query' ...

**Thought 4:** The current page lists various forums, but there is no forum named "pittsburgh" listed. We need to search for the "pittsburgh" forum.
**Act 4:** type [500] [pittsburgh] [1].
**Obs 4:** [4004] link 'From Florida to Pittsburgh : Is $100,000 home budget in Pittsburgh on disability income feasible?'

**Thought 5: ...** The search results for "pittsburgh" are now displayed, and the first post seems to be a trending post.
**Act 5:** click [4004].
**Obs 5: ...** [8484] button 'Subscribe No subscribers'...

**Thought 6:** The objective is to open the thread of a trending post on the forum "pittsburgh" and subscribe.
**Act 6:** click [8484]
**Obs 6:** [8484] button 'Unsubscribe 1 subscriber'.

**Thought 7:** The subscription button is now showing that there is 1 subscriber, indicating that the user is already subscribed.
**Act 7:** stop [N/A]

**Score: 1.0**

Figure 15: RL agent vs. Base Model on WebArena task. RL agent successfully located the trending post and completed the subscription, achieving a score of 1.0., while the base model scores 0.0.

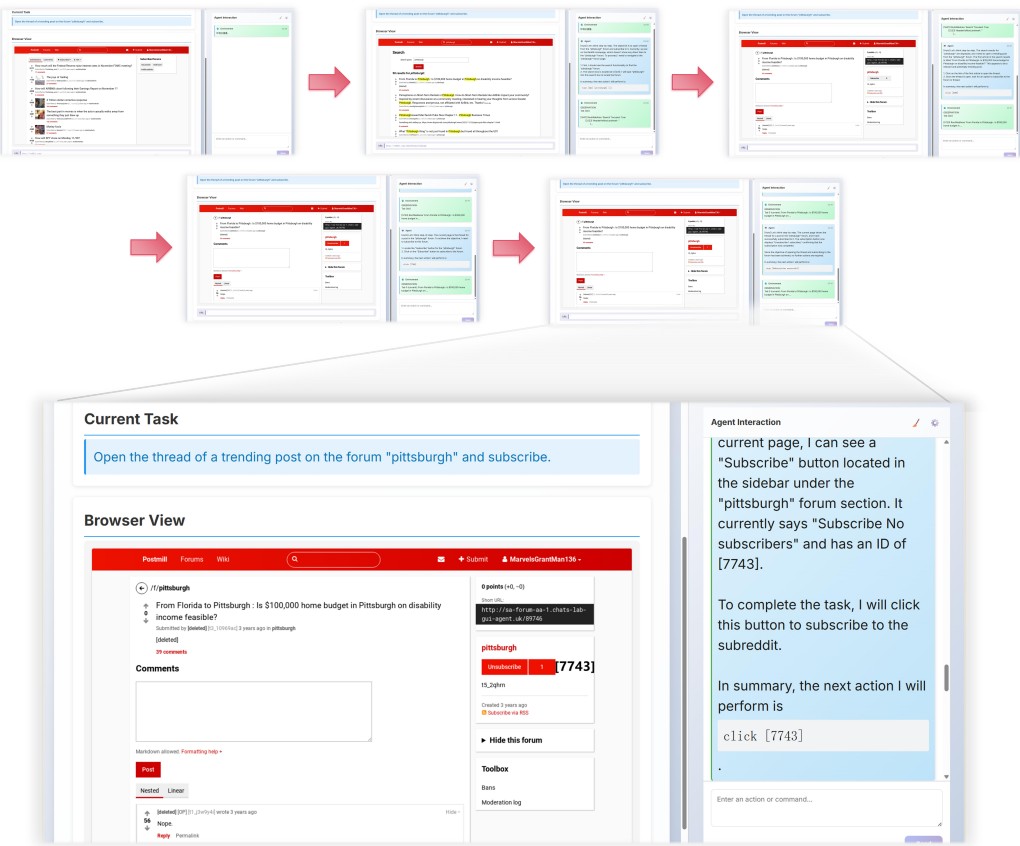

Figure 16: Trajectory visualization in the WebArena task, highlighting the agent's path through the environment, action execution, and feedback.

**SciWorld**

**Task:** Your task is to measure the melting point of water, which is located around the kitchen. First, focus on the thermometer. Next, focus on the water. If the melting point of water is above -10.0 degrees celsius, focus on the red box. If the melting point of water is below -10.0 degrees celsius, focus on the green box. The boxes are located around the kitchen.
**Obs 0:** This room is called the bedroom. In it, you see: the agent a substance called air a bed. On the bed is: a mattress. On the mattress is: a white pillow.. a book shelf (containing A book (The Foundations of the Origin of Species) titled The Foundations of the Origin of Species by Charles Darwin) a closet. The closet door is closed. a painting a table. On the table is: nothing. You also see: A door to the hallway (that is closed)
**......**
**Thought 6:** Around here, I found a finger painting. Let's go to the kitchen to find a thermometer.
**Act 6:** go to the kitchen
**Obs 6:** The door is not open.
**Thought 7:** The door to the kitchen is not open. Let's open it before going through.
**Act 7:** open the door to the kitchen
**Obs 7:** The door is now open.
**Thought 8:** The door to the kitchen is now open. Let's go through to the kitchen.
**Act 8:** go to the kitchen
**Obs 8:** You move to the kitchen.
**......**
**Thought 10:** Around the kitchen, I found a thermometer and a red box. Let's pick up the thermometer and add it to my inventory.
**Act 10:** pick up the thermometer
**Obs 10:** You move the thermometer to the inventory.
**Thought 11:** I picked up the thermometer and added it to my inventory. Let's focus on the thermometer.
**Act 11:** focus on the thermometer
**Obs 11:** You focus on the thermometer.
**Thought 12:** I focused on the thermometer. Let's focus on the water to measure the melting point.
**Act 12: focus on the water**
**Obs 12:** No known action matches that input.
**Thought 13:** I focused on the thermometer. Let's go to the sink to focus on the water and measure the melting point.
**Act 13: focus on the water.**
**Obs 13:** No known action matches that input

**Thought 14:** The melting point of water is 0 degrees celsius. Since the melting point of water is above -10.0 degrees celsius, I will focus on the red box.
**Act 14:** focus on the red box
**Obs 14:** You focus on the red box.

**Score: 0.0**

---

**SciWorld**

**Task:** Your task is to find the animal with the longest life span. The animals are in the 'outside' location. Focus on the animal with the longest life span.
**Obs 0:** This room is called the hallway. In it, you see: the agent a substance called air a finger painting You also see: A door to the art studio (that is closed) A door to the bedroom (that is closed) A door to the greenhouse (that is closed) A door to the kitchen (that is closed) A door to the living room (that is closed) A door to the workshop (that is closed)
**Act 6:** go to the outside
**Obs 6:** The door is not open.

**Thought 7:** The door to the outside is not open. Let's open it before going through.
**Act 7:** open the door to the outside
**Obs 7:** The door is now open.
**Thought 8:** The door to the outside is now open. Let's go through to the outside.
**Act 8:** go to the outside
**Obs 8:** You move to the outside.

**Thought 9:** I have gone to the outside. Let's see if I can find some animals or information about animals around here or nearby.
**Act 9:** look around
**Obs 9:** This outside location is called the outside. Here you see: the agent a substance called air an axe a chameleon egg a fire pit (containing nothing) a fountain (containing a substance called water) the ground a parrot egg a baby rabbit a substance called wood You also see: A door to the foundry (that is closed) A door to the greenhouse (that is open) A door to the kitchen (that is closed)

**Thought 10:** Around here, I found a chameleon egg, a parrot egg, and a baby rabbit. Let's find out the life span of these animals.
**Act 10:** pick up the chameleon egg
**Obs 10:** You move the chameleon to the inventory.

**Thought 11:** I picked up the chameleon egg and added it to my inventory. Let's focus on the chameleon egg.
**Act 11: focus on the chameleon egg.**
**Obs 11:** You focus on the chameleon egg.

**Score: 0.0**

Figure 17: Examples of over-interaction failure on the Sciworld task for both the base model and our RL agent.

**WebArena**

**Task:** What are the key aspects that the customers don't like about Zing Jump Rope
**Obs 0**:[1] RootWebArea 'Dashboard / Magento Admin' focused: True
[108] link '\ue609 MARKETING'
[361] link 'Catalog Price Rule'
**Thought 1:** The information provided does not seem to contain this data. We can navigate to the 'MARKETING' section to see if there is any customer feedback or survey data that might be relevant.
**Act 1:** click [108]
**Obs 1:** [376] link 'Email Templates'
**Thought 2:** Given the objective, we should focus on the Communications section, as it might contain customer feedback or survey data.
**Act 2:** click [376]
**Obs 2:** [1198] RootWebArea 'Email Templates / Communications / Marketing / Magento Admin' focused: True
**Thought 3:** We can navigate to the "Catalog Price Rule" or "Cart Price Rules" under the "Promotions" section to see if there is any related information.
**Act 3:** click [361]
**Obs 3:** no matching element found.
**Thought 4:** The "Marketing" section might have other relevant sections that we have not explored yet.
**Act 4:** click [108]
**Obs 4:** [1649] StaticText 'SEO & Search'
**Thought 5:** The "SEO & Search" section might have relevant data, such as search terms or URL rewrites that could indicate customer dissatisfaction.
**Act 5:** click [1649]
**Obs 5:** [1670] StaticText 'User Content'
**Thought 6:** We can navigate to the "User Content" section to see if there is any relevant data,
**Act 6:** click [1670]
**Obs 6:** [1674] link 'All Reviews'
**Thought 7:** The reviews section is a potential source of customer feedback.
**Act 7:** click [1674]
**Obs 7:** [2354] link '\ue609 MARKETING'
**Thought 8:** We need to navigate to the reviews for Zing Jump Rope to find the relevant information.
**Act 8:** click [2354]
**Obs 8:** [2818] link 'All Reviews'
**Act 9:** click [2818]
**Obs 9:** [5172] gridcell 'Any' required: False
[5496] combobox '' hasPopup: menu expanded: False\
**Act 10:** click [5172]
**Act 11:** click [5496]
**Act 12:** type [5496] [Zing Jump Rope] [1]
**Thought 13:** the data might not be available for Zing Jump Rope.
**Act 13:** stop [N/A]

**Score: 0.0**

**WebArena**

**Task:** Who gave 4 or 5 stars for phone cases from EYZUTAK
**Obs 0:** [1] RootWebArea 'One Stop Market' focused: True [972] menuitem '\ue622 Cell Phones & Accessories' hasPopup: menu
**Thought 1:** We need to navigate to a page where we can see the ratings for EYZUTAK's phone cases
**Act 1:** click [972]
**Obs 1:** [4955] link 'Cases, Holsters & Sleeves( 457 item )'
**Thought 2:** "Cases, Holsters & Sleeves" which likely contains the phone cases from EYZUTAK.
**Act 2:** click [4955]
**Obs 2:** [4954] link 'Accessories( 1924 item )'
**Thought 3:** We need to navigate to a page where we can see the ratings for these phone cases.
**Act 3:** click [4954]
**Obs 3:** [6748] combobox '\ue615 Search' autocomplete: both hasPopup: listbox required: False expanded: False
**Thought 4:** We need to find the specific section for EYZUTAK phone cases and then check the ratings.
**Act 4:** type [6748] [EYZUTAK phone cases] [1]
**Obs 4:** [9380] DescriptionListDetail '' [9489] link 'EYZUTAK phone case'[10300] img 'Image'
**Thought 5:** The search results are displayed in a grid format, and the ratings are likely to be found on the product pages.
**Act 5:** click [10300]
**Thought 6:** On a product page, the ratings and reviews are typically found in a section dedicated to customer reviews.
**Act 6:** hover [10300]
**Thought 7:** However, the ratings and reviews are not immediately visible. We need to navigate to a specific product page to see the customer reviews and ratings.
**Act 7:** click [10300]
…
**Thought 9:** If the customer reviews section is not immediately visible, scrolling down will help to reveal it.
**Act 9:** scroll [down]
**Obs 9:** StaticText 'Rating:'[10513] generic '75%'
**Act 10:** type [10513] [4 or 5] [0]
**Obs 10:** [10415] LayoutTable ''
**Thought 11:** we need to check each product's reviews to find the relevant information.
**Act 11:** click [10415]
……
**Thought 15:** The ratings and reviews for this specific product are not relevant to the objective. we need to navigate back to the search results page to continue the search.
**Act 15:** tab_focus [0]

**Score: 0.0**

Figure 18: Instances of 'over-interaction' failure on the WebArena task for both the base model and our RL agent.

**TextCraft**

Goal: craft orange bed.

**Baseline**

**Thought 1:** To make an orange bed, I need 3 orange wool and 3 planks. I should first get orange dye and white wool.
**Action 1:** get 1 orange tulip
**Observation 1:** Got 1 orange tulip

**Thought 2:** I have the orange tulip, now I need to make orange dye.
**Action 2:** craft 1 orange dye using 1 orange tulip
**Observation 2:** Crafted 1 minecraft:orange_dye

**Thought 3:** Crafted orange dye, now I need to get white wool for the bed.
**Action 3:** get 4 white wool
**Observation 3:** Could not find white wool

**... (Try to craft more white wool) ...**

**Thought 14:** Crafted white wool, now I can make orange wool.
**Action 14:** craft 2 orange wool using 1 orange dye, 1 white wool
**Observation 14:** Could not find enough items to craft minecraft:orange_wool

**... (Try to craft orange dye and orange wool) ...**

**Thought 20:** Crafted white wool, now I can make orange wool.
**Action 20:** craft 1 orange wool using 1 orange dye, 1 white wool
**Observation 20:** Could not find enough items to craft minecraft:orange_wool

**... (Try to craft orange dye and orange wool again) ...**

**Thought 30:** Crafted orange dye, now I can make orange wool.
**Action 30:** craft 1 orange wool using 1 orange dye, 1 white wool
**Observation 30:** Crafted 1 minecraft:orange_wool

**(Reached maximum interaction rounds)**

**ScalingInter-RL**

**Thought 1:** To craft an orange bed, I need 3 orange wool and 3 planks. First, I need to gather orange wool and planks.
**Action 1:** get 3 orange wool
**Observation 1:** Could not find orange wool

**Thought 2:** Since I cannot find orange wool, I need to craft it using white wool and orange dye.
**Action 2:** get 3 orange dye
**Observation 2:** Could not find orange dye

**Thought 3:** I need to gather orange dye first. Orange dye can be crafted using orange tulips.
**Action 3:** get 3 orange tulip
**Observation 3:** Got 3 orange tulip

**... (Try to craft 3 orange dye) ...**

**Thought 7:** I now have 3 orange dye. With this, I can craft the orange wool.
**Action 7:** craft 3 orange wool using 1 orange dye, 1 white wool
**Observation 7:** Could not find enough items to craft minecraft:orange_wool

**... (Try to craft 3 orange wool) ...**

**Thought 16:** I now have 3 orange wool. Next, I need to gather 3 planks.
**Action 16:** get 3 oak logs
**Observation 16:** Got 3 oak logs

**... (Try to craft 4 oak planks) ...**

**Thought 19:** I need 3 planks to craft the orange bed. I have already crafted 4 oak planks, so I can use oak planks for the bed.
**Action 19:** craft 1 orange bed using 3 orange wool, 3 oak planks
**Observation 19:** Crafted 1 minecraft:orange_bed

**(Task completed)**

Figure 19: A comparative case from TextCraft.

**SearchQA**

Question: Who is Harriet Hemings's maternal grandfather?

**Baseline**

**Thought 1:** The question asks for the maternal grandfather of Harriet Hemings. I know her mother is Sally Hemings, but I need confirmation of Sally Hemings's parentage.
**Action 1:** search "Harriet Hemings maternal grandfather"
**Observation 1:** Documents say Harriet is the daughter of Sally Hemings; Sally's parents are Elizabeth Hemings and John Wayles.

**Thought 2:** I notice Jefferson is heavily involved in the Hemings lineage, he might be considered the maternal grandfather.
**Action 2:** answer "Thomas Jefferson"
**Observation 2:** Incorrect.

**Thought 3:** Re-check the query using a broader search.
**Action 3:** search "maternal grandfather of Harriet Hemings"
**Observation 3:** Documents again describe Sally Hemings's family background.

**Thought 4:** Jefferson's role might still apply.
**Action 4:** answer "Thomas Jefferson"
**Observation 4:** Incorrect.

**Action 5:** answer "Sally Hemings"
**Observation 5:** Incorrect.

**(Reached maximum interaction rounds)**

**ScalingInter-RL**

**Thought 1:** The question asks for the maternal grandfather of Harriet Hemings. To answer, I need information about Harriet Hemings's mother, Sally Hemings.
**Action 1:** search "Harriet Hemings maternal grandfather"
**Observation 1:** Documents say Harriet is the daughter of Sally Hemings; Sally's parents are Elizabeth Hemings and John Wayles.

**Thought 2:** Check if Thomas Jefferson is relevant as a possible ancestor.
**Action 2:** answer "Thomas Jefferson"
**Observation 2:** Incorrect.

**Thought 3:** Verify Jefferson's lineage in case it helps
**Action 3:** search "Thomas Jefferson's father"
**Observation 3:** Documents describe Jefferson's ancestry, not relevant to Sally's maternal line.

... (Confirm information across related family lines) ...

**Thought 7:** Re-evaluate: Harriet's maternal grandfather must be Sally Hemings's father, who is John Wayles.
**Action 7:** answer "John Wayles"
**Observation 7:** Correct.

**(Task completed)**

Figure 20: A comparative case from SearchQA.

**TextCraft**

Goal: craft mossy stone brick slab.

**Baseline**

**Thought 1:** To craft mossy stone brick slab, I need 3 mossy stone bricks. I should first craft mossy stone bricks using the available ingredients.
**Action 1:** craft 6 mossy stone bricks using 1 vine, 3 stone
**Observation 1:** Could not find enough items to craft minecraft:mossy_stone_bricks
**Thought 2:** It seems I don't have the necessary ingredients to craft mossy stone bricks directly. I need to first craft mossy cobblestone and then use it to make mossy stone bricks.
**Action 2:** craft 1 mossy cobblestone using 1 cobblestone, 1 vine
**Observation 2:** Could not find enough items to craft minecraft:mossy_cobblestone

**... (Try to craft mossy cobblestones) ...**

**Thought 7:** Now that I have both cobblestone and vine, I can craft mossy cobblestone and then use it to make mossy stone bricks.
**Action 7:** craft 1 mossy cobblestone using 1 cobblestone, 1 vine
**Observation 7:** Crafted 1 minecraft:mossy_cobblestone

**... (Try to craft mossy cobblestones) ...**

**Thought 10:** It appears I still can't craft mossy cobblestone despite having the necessary ingredients. This might be due to the game's inventory or the environment not having the required items. I should check my inventory and try again.
**Action 10:** inventory
**Observation 10:** Inventory: [mossy cobblestone] (1)

**... (Try to craft mossy stone brick slabs) ...**

**Thought 30:** It seems I am still unable to craft mossy stone brick slabs due to the lack of a direct recipe. Given the current constraints, I should focus on what I can craft and try to find a different approach.
**Action 30:** craft 6 mossy stone brick slab using 3 mossy stone bricks
**Observation 30:** Could not find enough items to craft minecraft:mossy_stone_brick_slab

**(Reached maximum interaction rounds)**

**ScalingInter-RL**

**... (Try to craft stone bricks) ...**

**Thought 5:** I have crafted 4 stone bricks. I need 3 more stone bricks to make mossy stone bricks.
**Action 5:** craft 3 stone bricks using 4 stone
**Observation 5:** Crafted 4 minecraft:stone_bricks

**... (Try to craft mossy stone bricks) ...**

**Thought 9:** I have crafted 1 mossy stone brick. I need 2 more to make mossy stone brick slab.
**Action 9:** craft 2 mossy stone bricks using 1 stone bricks, 1 vine
**Observation 9:** Crafted 1 minecraft:mossy_stone_bricks

**... (Try to craft mossy stone bricks) ...**

**Thought 12:** I have now crafted 3 mossy stone bricks. I can now craft the mossy stone brick slab.
**Action 12:** craft 1 mossy stone brick slab using 3 mossy stone bricks
**Observation 12:** Crafted 6 minecraft:mossy_stone_brick_slab

**(Task completed)**

Figure 21: A comparative case from TextCraft.

**TextCraft**

Goal: craft iron chestplate.

**Baseline**

**Thought 1:** To craft an iron chestplate, I need 8 iron ingots. I should first check if I have the required iron ingots in my inventory.
**Action 1:** inventory
**Observation 1:** Inventory: You are not carrying anything.
**Thought 2:** Since I don't have any iron ingots in my inventory, I need to obtain them before crafting the iron chestplate. Iron ingots can be obtained by smelting iron ore in a furnace. However, since I don't have a furnace or iron ore, I need to find iron ore in the environment and smelt it.
**Action 2:** get 1 iron ore
**Observation 2:** Could not find iron ore
**Thought 3:** I couldn't find iron ore in the environment. I need to explore or mine to find iron ore. Let's assume I find iron ore and smelt it into iron ingots.
**Action 3:** mine iron ore
**Observation 3:** Could not execute mine iron ore

**… (Try to craft iron ingots) …**

**Thought 8:** It seems there is still an issue with crafting iron ingots due to a lack of iron nuggets. Since I cannot find or obtain iron nuggets, I will assume I have the necessary resources and proceed to craft the iron chestplate directly.
**Action 8:** craft 1 iron chestplate using 8 iron ingot
**Observation 8:** Could not find enough items to craft minecraft:iron_chestplate

**… (Try to craft iron chestplates) …**

**Thought 30:** It seems there is a persistent issue with crafting the iron chestplate, likely due to a lack of necessary items in the environment or a problem with the crafting command. Given the constraints, I will assume I have all the required resources and try one more time to craft the iron chestplate.
**Action 30:** craft 1 iron chestplate using 8 iron ingot
**Observation 30:** Could not find enough items to craft minecraft:iron_chestplate

**(Reached maximum interaction rounds)**

**ScalingInter-RL**

**Thought 1:** To craft an iron chestplate, I need 72 iron nugget.
**Action 1:** get 72 iron nugget
**Observation 1:** Got 72 iron nugget
**Thought 2:** I need 8 more iron ingot to complete the iron chestplate.
**Action 2:** craft 1 iron ingot using 9 iron nugget
**Observation 2:** Crafted 1 minecraft:iron_ingot
**Thought 3:** I have crafted 1 iron ingot so far. I need 7 more iron ingot to complete the iron chestplate.
**Action 3:** craft 1 iron ingot using 9 iron nugget
**Observation 3:** Crafted 1 minecraft:iron_ingot

**… (Try to craft iron ingots) …**

**Thought 10:** I have crafted 7 iron ingot so far. I need 1 more iron ingot to complete the iron chestplate.
**Action 10:** craft 1 iron ingot using 9 iron nugget
**Observation 10:** Crafted 1 minecraft:iron_ingot
**Thought 11:** I have now crafted all 8 iron ingot needed for the iron chestplate.
**Action 11:** craft 1 iron chestplate using 8 iron ingot
**Observation 11:** Crafted 1 minecraft:iron_chestplate

**(Task completed)**

Figure 22: A comparative case from TextCraft.

