# OpenReview forum: "AgentGym-RL: An Open-Source Framework to Train LLM Agents for Long-Horizon Decision Making via Multi-Turn RL"
_ICLR.cc/2026/Conference — ICLR 2026 Oral_

### Official Review · Reviewer_yP1T · 2025-10-26

**Soundness:** 3
**Presentation:** 3
**Contribution:** 2
**Rating:** 6
**Confidence:** 3

**Summary:**

This paper presents AgentGym-RL, an open-source framework for training LLM agents via RL in multi-turn decision-making tasks. The main contributions include: a modular framework supporting diverse environments and mainstream RL algorithms, and a staged training approach named ScalingInter-RL that progressively increases interaction horizons to achieve stable long-horizon RL training. The authors demonstrate that 7B models trained with their method achieve performance comparable to or surpassing commercial models like OpenAI o3 and Gemini-2.5-Pro across 27 tasks.

**Strengths:**

1. The motivation is clear and meaningful. The paper tackles the genuine problem of lacking unified RL frameworks for training LLM agents across diverse environments, which is important for the community.
2. As an open-source framework, this work provides good engineering with modular design, supporting multiple RL algorithm for diverse environments as well as various practical features.
3. The framework demonstrates strong empirical performance. It achieves performance matching or surpassing much larger commercial models with 7B parameters, which is impressive, especially with the 33.65 point average improvement.
4. The evaluation is comprehensive, which tests across 5 scenarios with 27 tasks. Such evaluation provides good coverage of different agent capabilities.

**Weaknesses:**

1. The algorithmic novelty is limited, although the effectiveness is acknowledged. ScalingInter-RL is essentially curriculum learning applied to interaction horizons. This is a straightforward adaptation rather than a novel algorithmic contribution. The progressive scaling from short to long horizons is intuitive and incremental, which lacks theoretical justification.
2. Given that the key to ScalingInter-RL is the curriculum scaling of interaction turns, there is no ablation on the curriculum schedule. Why these specific horizon increments?
3. I could not find quantitative report and analysis of computational costs or training efficiency.

**Questions:**

1. Could the authors provide a curriculum sensitivity study? What happens with different progression rates or starting horizons of interactions? This seems critical but is unexplored.
2. What are the computational costs? The paper doesn't discuss training time, GPU hours, or cost comparisons with baseline approaches, making it hard to assess practical viability.
3. Are there separate studies investigating whether the improvement mainly comes from the framework or from the ScalingInter-RL method? Would ScalingInter-RL work in other frameworks? Would the framework benefit from other training approaches?

---

> ### Author Response · Authors · 2025-11-20
> **Response to Reviewer yP1T.**
>
> 1. **Question about the ablation study of ScalingInter-RL.**
>    - Thank you very much for your suggestion. In  **Point #1 of the General Response** , we conduct a detailed, multi-dimensional ablation study of ScalingInter-RL (initial number of interaction steps, the increment size (δh), and the frequency of phase transitions (Δ)), showing that it is not sensitive to these hyperparameters.
>    - Meanwhile,  **in Points #2, #3, and #4 of the General Response** , we present several additional experimental designs. If you are interested, please refer to them; they show that more fine-grained designs can bring performance improvements.
>    - We will follow your suggestion and incorporate these results into the manuscript. Thank you again!
> 2. **Question about the contribution of ScalingInter-RL.**
>    - Thank you very much for your feedback! It made us realize that we should better clarify where our contributions come from. As you pointed out, our contributions lie in our clear motivation, the proposed framework, and the demonstrated effectiveness of the method; we also hope to provide a solid foundation and some insights for the open-source community.
>    - We introduce the idea of *Scaling Interaction* and argue that it is key to building effective LLM-based agents. We identify several limitations of vanilla method like training instability and exploding gradients, as shown in  **the Figure 11 of the revised manuscript** :
>    - In addition, we also analyzed the error patterns of the vanilla method (see  **Q7 of the Response to Reviewer gQCC** ).
>    - We conduct an in-depth analysis, and introduce our method ScalingInter-RL. Its effectiveness is validated through extensive and comprehensive experiments.
>    - Thank you again for your question!
> 3. **Question about the analysis of resources and efficiency.**
>    - Thank you very much for your question, which drew our attention to the efficiency of ScalingInter-RL. We provide this analysis in **Point #8 of the General Response (Figure 10 in the revised manuscript)**. Inspired by [1], by presenting the training reward, the cumulative number of interaction steps, and the total training time, we show that our method achieves strong performance with relatively little training time and fewer interaction steps.
> 4. **Question about the analysis of methods from different frameworks.**
>    - Thank you very much for your question. First, in **Point #5 of the General Response**, we conduct real-world experiments where training uses the Google Search API and evaluation is performed on the latest benchmarks (GAIA [2], BrowseComp [3], and xbench-ds [4]). The experimental results show that our method significantly outperforms the baselines, indicating that it also performs well in real-world environments.
>    - In addition, we validated our method with different RL algorithms ( **Point #4 of the General Response** ), including PPO and REINFORCE++, demonstrating the applicability of our approach across algorithms.
>    - Following your suggestion, we also adopt the RAGEN [5] framework to further verify the effectiveness of our method. The corresponding experiments are reported in  **Point #6 of the General Response** , where we find that ScalingInter-RL achieves quite strong performance across different frameworks.
>
> Once again, thank you for your valuable comments! We hope our responses address your concerns; if you have any further questions, please feel free to let us know and we will do our best to respond. If you find our replies satisfactory, we kindly ask that you consider updating your score and confidence accordingly.
>
> [1] Skywork Open Reasoner 1 Technical Report
>
> [2] GAIA: A BENCHMARK FOR GENERAL AI ASSISTANTS
>
> [3] BrowseComp: A Simple Yet Challenging Benchmark for Browsing Agents
>
> [4] https://xbench.org/
>
> [5] RAGEN: Understanding Self-Evolution in LLM Agents via Multi-Turn Reinforcement Learning

---

> > ### Comment · Reviewer_yP1T · 2025-11-26
> >
> > I appreciate the authors' efforts in providing the previously missing ablation on initial number of interaction rounds, the stage transition frequency, and the interaction interval. The result showing insensitivities actually highlights the robustness of the proposed method, which is good. I thank the author for conducting a systematic efficiency analysis, where the result demonstrates strong performance. It is also great to see that the authors further validated their method with additional RL algorithms and different frameworks. The rebuttal has successfully addressed my questions and concerns, and I am happy to raise my score.

---

> > > ### Author Response · Authors · 2025-11-26
> > > **Thank you!**
> > >
> > > We are very glad to hear that our rebuttal has addressed your questions and concerns. Thank you for your thoughtful feedback, your recognition of our work. We truly appreciate your time and consideration, and we will continue to improve our work!

---

### Official Review · Reviewer_gQCC · 2025-10-27

**Soundness:** 3
**Presentation:** 3
**Contribution:** 2
**Rating:** 6
**Confidence:** 3

**Summary:**

This paper introduces AgentGym-RL, a modular open-source framework for training Large Language Model (LLM) agents on complex, multi-turn decision-making tasks using Reinforcement Learning (RL). The authors identify a key gap in the current open-source community: the lack of a unified platform for training agents from scratch in diverse and realistic environments. AgentGym-RL addresses this by providing a decoupled architecture—comprising environment, agent, and training modules—that supports various scenarios (e.g., WebArena, Deep Search, SciWorld) and mainstream RL algorithms.
To tackle the training instability common in long-horizon tasks, the paper proposes a curriculum learning-based strategy named ScalingInter-RL. This method begins training with shorter interaction episodes to establish a foundational policy and then progressively increases the episode length. This approach enables stable training and deeper exploration. Extensive experiments across 27 tasks demonstrate that a 7B parameter model trained with this method can match or even surpass the performance of much larger proprietary models like GPT-4 and Gemini Pro, suggesting that scaling interaction can be more effective than simply scaling model parameters.

**Strengths:**

1. The development of a unified, open-source RL framework for LLM agents is a valuable contribution to the research community. It directly addresses a practical need and has the potential to significantly lower the barrier to entry for agent research, thereby fostering reproducibility and continued innovation.
2. ScalingInter-RL is an intuitive yet powerful method that effectively stabilizes long-horizon RL training. The paper provides a clear motivation for this curriculum-based strategy (as illustrated in Figure 4), showing how it adeptly balances the trade-off between exploration depth and training stability. Its effectiveness is validated by solid experimental results.
3. The experimental design is a major highlight of this work. The evaluation spans five diverse and challenging environments and comprehensively compares the proposed model against a wide range of strong open-source and state-of-the-art proprietary models. This thorough evaluation lends significant credibility to the paper's conclusions.

**Weaknesses:**

1. Limited Analysis of the ScalingInter-RL Schedule: While the method is shown to be effective, the paper offers limited analysis of the curriculum schedule itself. The model's final performance is likely sensitive to hyperparameters such as the initial number of interaction steps, the increment size (δh), and the frequency of phase transitions (Δ). A sensitivity analysis or ablation study on these hyperparameters would provide deeper insights into the method's robustness and offer guidance for its application in new domains, thereby strengthening the paper's claims.
2. Rigor of the Exploration-Exploitation Argument: The paper states that by initially limiting the interaction horizon, the agent focuses more on exploitation, and as the horizon expands, it is encouraged to explore more deeply. This description could be more rigorous. Even in the early stages, exploration is necessary to discover better trajectories for policy improvement. It would be beneficial to contrast this with classic exploration techniques in RL, such as curiosity-driven methods like RND or NGU, to better contextualize the exploration-exploitation trade-off within ScalingInter-RL.

**Questions:**

1. Sensitivity of ScalingInter-RL: Could the authors provide more details on the robustness of the ScalingInter-RL method? For instance, how does performance vary with different curriculum schedules (e.g., faster/slower expansion of the interaction horizon, different starting points)? A related ablation study would be highly informative.
2. Instability in Long-Horizon Training: Could you elaborate on why training is inherently more unstable in long-horizon settings? Providing related experimental or theoretical analysis would be a valuable contribution to the community.
3. Overhead of HTTP Communication: Is there an analysis of the time overhead associated with the environment client communicating with the environment server via HTTP? A comparison with a fully collocated setup would be very insightful.
4. Interaction with Different RL Algorithms: The paper compares GRPO and REINFORCE++, but the main benefits of ScalingInter-RL appear to be demonstrated with GRPO. It would be beneficial to know if the stability and performance gains from ScalingInter-RL are consistent across other RL algorithms, such as PPO. This would help confirm whether the method is a general strategy for stabilizing long-horizon training, rather than a technique tightly coupled with the specific mechanics of GRPO.
5. Computational Overhead: How do the training time and computational costs of ScalingInter-RL compare to a baseline with a fixed, long horizon? By avoiding unstable training phases, does this staged approach lead to more efficient use of computational resources, potentially achieving a high-performance policy with faster overall convergence?
6. Comparative Failure Case Analysis: The case studies in Appendix F are very helpful. It would be even more compelling to see a direct comparison of failure modes on a complex task between a baseline agent (trained with a fixed short horizon) and the ScalingInter-RL agent. Does ScalingInter-RL specifically help mitigate certain types of errors, such as repetitive loops or failure to explore deeper parts of the state space?

---

> ### Author Response · Authors · 2025-11-20
> **Response to Reviewer gQCC. Part [1/2]**
>
> Thank you very much for your recognition of our work and for your valuable suggestions. Our responses are as follows:
>
> 1. **Question about the ablation study on ScalingInter-RL.**
>    - Thank you very much for this suggestion. In  **Point #1 of the General Response** , we have conducted a detailed multi-dimensional ablation study of ScalingInter-RL (the initial number of interaction steps, the increment size (δh), and the frequency of phase transitions (Δ)), showing that it is not sensitive to these hyperparameters. As you noted, this helps strengthen the paper’s claims. We will follow your suggestion and incorporate these results into the manuscript. Thank you again!
> 2. **Question about exploration and exploitation in LLM agents, and the use of classical exploration techniques from RL.**
>    - Thank you very much for your question and suggestions, which made us realize that further clarification is needed. In our work, exploration for LLM agents mainly refers to the diversity of actions and the depth of trajectories. We hope that under a diverse set of queries, the model can achieve good performance through sufficiently deep exploration. This metric is indeed difficult to quantify, but we find that if we avoid extremely long interactions at the early stage and instead let the model first reliably solve simple problems, and only later encourage long-horizon exploration, the training dynamics become more stable.
>    - Following your suggestion, we also experimented with the NGU exploration technique from classical RL, which applies the RND mechanism you mentioned. Specifically, we use an embedding model to map environment observations into vectors, and then compute state novelty to construct an intrinsic reward. Concretely, within each episode, we compute the distance between the current state and its nearest neighbour in the list of previously visited state vectors to obtain within-episode novelty, with repeated states in the same episode receive almost no reward. In parallel, we use RND to measure novelty from a life-long training perspective over the entire training process, i.e., the rarer a state, the larger the prediction error and the higher the reward; repeated visits naturally reduce the reward. We then train the agent using a weighted combination of the two intrinsic rewards (within-episode and across-episode) and the external environment reward. The comparative training results are as follows:
>       | Model         | BabyAI |
>       | ------------- | ------ |
>       | Base Model    | 69.7   |
>       | ScaleInter-RL | 96.7   |
>       | NGU           | 88.9   |
>    - **During training, we found that the NGU method generally encourages the model to perform a large amount of exploration, sometimes even leading to over-exploration and excessive reasoning.** This caused multiple sudden drops in validation accuracy, and the final performance was inferior to ScalingInter-RL. In the future, we look forward to experimenting with more classical RL exploration techniques and making them contribute to performance improvements in a stable manner.
> 3. **Question about the instability of Long-Horizon Training?**
>    - From a theoretical perspective, for a given query, an LLM agent performs multiple rounds of interaction, where each round is a ReAct step (Reasoning + Action). Each step consists of multiple tokens, so the entire interaction produces a large volume of tokens, which makes the task very challenging for an agent that lacks sufficient environmental context and basic capabilities.
>    - We conducted an empirical analysis, as shown in **the Figure 11 in Appendix H in the revised manusccipt**.
>    - **We found that, in terms of training reward, long-horizon training tends to collapse, mainly due to the uncertainty introduced by long-horizon interactions and operations.** At the same time, we observed that this phenomenon is often accompanied by abnormal gradients, where the **grad norm exhibits multiple spikes**, which empirically confirms its instability.
> 4. **Question about the overhead of HTTP communication.**
>    - Thank you very much for this question, which made us realize that we had not clarified this point and prompted us to provide a more insightful analysis. Following your suggestion, we conducted an analysis in  **Point #7 of the General Response**. The experimental results show that the overhead of HTTP communication during interaction is very small, accounting for only about 1% of the total.

---

> ### Author Response · Authors · 2025-11-20
> **Response to Reviewer gQCC. Part [2/2]**
>
> 5. **Question about experiments with more RL algorithms.**
>    - Thank you very much for this suggestion, which reminded us that we can use different algorithms to validate the effectiveness of ScalingInter-RL. Following your suggestion, we carried out experiments in  **Point #4 of the General Response**. The results show that, although there are indeed performance differences across algorithms, ScalingInter-RL consistently brings performance gains to different algorithms, confirming that it is a general strategy.
> 6. **Question about the efficiency of the ScalingInter-RL method.**
>    - Thank you very much for this suggestion, which drew our attention to the efficiency of ScalingInter-RL. We performed an analysis in **Point #8 of the General Response (Figure 10 in the revised manuscript)**. Inspired by [1], we present the training reward, the cumulative number of interaction rounds, and the total training time, and we verify that our method achieves strong performance with relatively few training rounds and less training time.
> 7. **Question about case study of short-turn-RL agent and ScalingInter-RL agent.**
>    - Thank you very much for your suggestion. Accordingly, we conducted an analysis on the search task. On the test set, we performed an error-type analysis and used GPT-5 to evaluate the likelihood that a trajectory contains the following five patterns (a single trajectory may exhibit multiple error types):
>
>       * Constraint Violation (CV): Agent violates explicit constraints or rules of the environment.
>       * Format Following (FF): Agent fails to follow required output formats or protocols.
>       * Goal Misalignment (GM): Agent's actions diverge from the stated objective.
>       * Repetitive Loop (RL): Agent gets stuck in repetitive, non-progressive behavior patterns.
>       * Shallow Exploration (SE): Agent fails to adequately explore the problem space or consider alternatives.
>    - Our experimental results show that both the baselines and ScalingInter-RL handle the first two types of errors well. **As you mentioned, short-horizon RL typically does not exhibit many Repetitive Loop phenomena, but it often suffers from Shallow Exploration. In contrast, long-horizon RL is more likely to fall into over-exploration and deviate from the original goal.**
>
>       | Method           | CV   | FF   | GM   | RL   | SE   |
>       | ---------------- | ---- | ---- | ---- | ---- | ---- |
>       | Base Model       | 0.31 | 0.46 | 0.01 | 0.44 | 0.32 |
>       | Short Horizon RL | 0.02 | 0.05 | 0.01 | 0.31 | 0.59 |
>       | Long Horizon RL  | 0.05 | 0.06 | 0.1  | 0.58 | 0.47 |
>       | ScalingInter-RL  | 0.02 | 0.06 | 0.01 | 0.41 | 0.45 |
>    - We show two cases in **Figure 19 and Figure 20 of the revised manuscript**:
>
>       * The first (**Figure 19 of the revised manuscript**) is a case from the TextCraft environment. **The short-horizon agent fails because it uses incorrect quantities (violating crafting constraints) and drifts from the goal of crafting an orange bed.** In contrast, our agent succeeds by adhering to constraints and  **managing resources strategically**: it proactively checks inventory, adaptively explores alternatives (e.g., crafting intermediate items), and efficiently acquires materials to resolve shortages. This showcases stronger long-term planning and underscores the effectiveness of our method.
>       * The second (**Figure 20 of the revised manuscript**) is a case from SearchQA environment. **The short-horizon agent fails because it only performs superficial searches.** When encountering misleading information (such as the references to the Jefferson family), it does not conduct further verification or cross-check the family lineage. **ScalingInter-RL agent, on the other hand, actively adjusts its strategy after making an error.** Through multiple turns of search, it verifies hypotheses, filters out irrelevant information, and further traces the parentage of Sally Hemings. It gradually narrows the search scope step by step, and ultimately identifies the key genealogical fact (that Sally Hemings’s father is John Wayles), allowing it to arrive at the correct answer.
>
> We once again thank you for your valuable comments!  If you have any further questions, please feel free to let us know and we will do our utmost to respond. If you find our replies satisfactory, we kindly ask that you consider updating your score and confidence accordingly.
>
> [1] Skywork Open Reasoner 1 Technical Report

---

> > ### Comment · Reviewer_gQCC · 2025-11-24
> > **Official Comment by Reviewer gQCC**
> >
> > I thank the authors for their comprehensive rebuttal. The new analyses have addressed my primary concerns and have strengthened the manuscript. My one remaining suggestion pertains to the concept of "over-exploration and excessive reasoning." I believe a more in-depth analysis of this phenomenon would be highly insightful. Therefore, I would encourage the authors to incorporate a deeper dissection of this into the final manuscript.

---

> > > ### Author Response · Authors · 2025-11-25
> > > **New Response to Reviewer gQCC.**
> > >
> > > Thank you very much for joining the discussion and sharing your suggestion! Based on your suggestion, we conducted a new analysis!
> > >
> > > First, **over exploration** refers to cases where the agent's behavior deviates from the intended goal and explores things irrelevant to reaching the target state. In addition, when agents are close to achieving the goal, they may switch to pursuing other unrelated objectives, which can lead to failure. **Excessive reasoning**, on the other hand, refers to situations where the agent spends more effort on internal reasoning rather than interacting with the environment to obtain more information; it may even speculate about or fabricate the environment's responses. This is inconsistent with our original motivation for **scaling interactions**.
> > >
> > > We ran experiments and, as in our previous error analysis, used GPT-5 as an LLM-based judge to perform the categorization. The results are shown in the table below. We find that when the window is smaller—i.e., under **Short-Horizon RL** —the model tends to think more, and thus exhibits more **excessive reasoning**. Under **Long-Horizon RL**, the model typically explores a lot, but much of this exploration is not necessarily useful; a substantial portion is redundant, as illustrated in Figure 4 of the manuscript, which leads to a notable drop in performance.
> > >
> > > | Method           | Over Exploration | Excessive Reasoning |
> > > | ---------------- | ---------------- | ------------------- |
> > > | Base Model       | 0.05             | 0.36                |
> > > | Short Horizon RL | 0.03             | 0.42                |
> > > | Long Horizon RL  | 0.10             | 0.39                |
> > > | ScalingInter-RL  | 0.02             | 0.34                |
> > >
> > > - Meanwhile, we also present two representative cases in Figures 21 and 22 of the revised manuscript.
> > >   - In Figure 21, the long-horizon agent failed because it did not follow the task objective and directly craft mossy stone bricks. Instead, it attempted an action less relevant to the goal—crafting mossy cobblestones—which actually moved it farther away from completing the target task. In contrast, our agent maintained goal-relevant exploration, correctly found the path to craft mossy stone bricks, and thus completed the task more efficiently. This case suggests that the long-horizon agent may suffer from over-exploration, engaging in redundant exploration in the environment while neglecting goal-directed progress.
> > >   - In Figure 22, the short-horizon agent failed because it did not explore the environment appropriately and sufficiently. Instead, it performed excessive internal reasoning and incorrectly assumed it needed to obtain iron ingots by mining iron ore, even though this was not required by the instructions at the current difficulty level. As a result, it overlooked a more direct way to obtain the ingredients needed to craft iron chestplates. In contrast, our agent explored the environment correctly and sufficiently, recognized that it could directly obtain the required ingredients from the environment, and—based on its exploration results—collected iron nuggets, then crafted iron ingots, and completed the task. This case indicates that the short-horizon agent may face an excessive reasoning issue: it relies too heavily on internal reasoning, makes incorrect assumptions about the environment state, and fails to interact with the environment to explore.
> > >
> > > Thank you again for your feedback—we will try our best to improve the manuscript accordingly. We hope we've addressed your concerns and strengthened your confidence in our paper, and we would greatly appreciate it if you could update your score.

---

> > > > ### Comment · Reviewer_gQCC · 2025-11-26
> > > > **Official Comment by Reviewer gQCC**
> > > >
> > > > I appreciate the authors' additional analysis. While the provided breakdown is largely quantitative, a deeper discussion on the underlying mechanisms behind these phenomena would be highly insightful for the community. Nevertheless, the rebuttal has clarified my primary concerns. I have updated my contribution score accordingly.

---

> > > > > ### Author Response · Authors · 2025-12-02
> > > > > **Thank you!**
> > > > >
> > > > > Dear Reviewer gQCC,
> > > > >
> > > > > We sincerely appreciate your valuable feedback, thoughtful engagement throughout the rebuttal phase, and your positive reconsideration of our work. We will continue to further improve our work!
> > > > >
> > > > > Best regards,
> > > > >
> > > > > Authors of ICLR 2026 Submission 20092

---

### Official Review · Reviewer_odrg · 2025-10-30

**Soundness:** 3
**Presentation:** 3
**Contribution:** 2
**Rating:** 6
**Confidence:** 3

**Summary:**

This paper introduces AgentGym-RL, a novel and unified reinforcement learning framework designed for training LLM agents to perform long-horizon, multi-turn interactive decision-making. To address the challenge of balancing exploration and exploitation and to ensure stable optimization, the authors propose ScalingInter-RL, a method that progressively scales the agent-environment interaction horizon during training. Extensive experiments demonstrate that the framework and method deliver significant and consistent performance gains.

**Strengths:**

- 1.The introduction of AgentGym-RL, the first modular, flexible, and end-to-end reinforcement learning framework specifically designed for training LLM agents in multi-turn, long-horizon decision-making across diverse real-world environments, filling a critical gap in the field.

- 2.The proposed ScalingInter-RL method progressively increases the interaction horizon during training. This innovative approach effectively balances exploration and exploitation, leading to more stable RL optimization and enabling agents to develop richer behaviors and strategies.

- 3.Extensive experiments demonstrate that the framework and method significantly enhance the capabilities of open-source models.

**Weaknesses:**

- 1. The proposed framework is built upon existing open-source projects like AgentGym and veRL. Its main contributions are engineering-oriented—adding diverse environments, integrating existing RL algorithms, and improving scalability and stability. These engineering improvements are certainly valuable and can significantly enhance usability and robustness; however, from a research perspective, the work represents a continuation and consolidation of existing efforts rather than a fundamentally novel contribution.

- 2. The ScalingInter-RL method lacks detailed specification on how initial-stage learning is concretely guided. Crucially, it omits the design of reward functions or subtasks for short-horizon phases, making it unclear how the agent acquires foundational skills before scaling up exploration.

- 3. The demonstrated success is more pronounced in structured, simulated environments with clear rules, while improvements in noisy, real-world settings are modest. This suggests the framework's effectiveness may be closely tied to environments that align well with its curriculum-based scaling approach.

**Questions:**

1. **On the specifics of early-stage curriculum learning**: The ScalingInter-RL method emphasizes exploitation by initially restricting interaction turns. Could you detail how the agent's learning is concretely guided during these short-horizon phases? Specifically, were task-specific dense reward functions or explicit subtasks designed to ensure the acquisition of foundational skills, rather than just learning to avoid quick failure? Furthermore, was this design implemented through a universal framework applicable across all environments, or was it individually tailored for each distinct task type?

- 2. **On generalization to more challenging RL tasks**: The experiments demonstrate strong performance in structured, language-mediated environments. How might the AgentGym-RL framework and the ScalingInter-RL approach be adapted to classic RL challenges that involve high-dimensional continuous state-action spaces?

---

> ### Author Response · Authors · 2025-11-20
> **Response to Reviewer odrg.**
>
> Thank you very much for your recognition of our work and for your valuable suggestions. Our responses are as follows:
>
> 1. **Question about the detailed specification of ScalingInter-RL at the initial stage.**
>    - In fact, we did not design a highly sophisticated reward function, nor did we introduce additional designs for data sampling, in order to keep the overall system simple. To further address this suggestion, we adopted several additional new approaches (see  **Point #3 in the General Response** ).
>    - We found that making extra designs (e.g., modifying the training data or the reward function) does influence the final performance of the model. In particular, constructing the training data based on difficulty tends to work reasonably well, but it introduces more computational overhead. On the other hand, our original experiments *without* such specialized designs already perform quite well.
>    - Thank you again for this suggestion!
> 2. **Question about our contributions.**
>    - Thank you very much for your feedback; it made us realize that we should clarify the nature and source of our contributions. Our work mainly lies in proposing a unified and flexible framework for the community, conducting extensive experiments to establish a solid foundation, and further improving LLM agent performance through several new and insightful methods.
>    - **We introduce the idea of *Scaling Interaction* and argue that it is key to building effective LLM-based agents. The decoupled design of our framework, its implementation, and its closer alignment with the needs of the agent community are important innovations.** The methodological innovations and the corresponding in-depth analyses provide valuable insights and enable stronger results on long-horizon interactive agents. Therefore, we believe that, beyond the engineering effort, our work indeed makes a meaningful contribution to the community.
>    - Thank you again; we will discuss this point more clearly in future versions of the manuscript.
> 3. **Question about the applicability in continuous high-dimensional spaces or real-world environments, i.e., whether our curriculum-based method is coupled to simulated environments.**
>    - In  **Point #1 of the General Response** , we conducted detailed ablation studies on ScalingInter-RL and showed that it is not sensitive to hyperparameters.
>    - In addition, we ran experiments in realistic scenarios and tasks (see  **Point #5 of the General Response** ), and evaluated on the latest benchmarks (GAIA[1], BrowseComp[2], and xbench-ds[3]). The results show that our method significantly outperforms the baselines. This indicates that our approach also works well in real environments and is not coupled to simulated environments.
>
> Once again, we sincerely thank you for your valuable comments! We hope our responses address your concerns. If you have any further questions, please do not hesitate to let us know—we will do our best to respond. If you find our replies satisfactory, we kindly ask you to consider adjusting your score and confidence accordingly.
>
> [1] GAIA: A BENCHMARK FOR GENERAL AI ASSISTANTS
>
> [2] BrowseComp: A Simple Yet Challenging Benchmark for Browsing Agents
>
> [3] https://xbench.org/

---

> > ### Author Response · Authors · 2025-11-28
> > **Invitation to discussion**
> >
> > Dear Reviewer odrg,
> >
> > We would like to thank you again for your thoughtful and constructive feedback, and we appreciate your recognition of the strengths of our paper. During the rebuttal stage, we conducted additional analyses and provided detailed clarifications to directly address your concerns. The key points are summarized below:
> >
> > - We provided further clarification about our contributions.
> > - We provided detailed specification of ScalingInter-RL at the initial stage.
> > - We performed detailed ablation studies on ScalingInter-RL to show that it is not sensitive to hyperparameters.
> > - We ran experiments in realistic scenarios and tasks and evaluated on the latest benchmarks (GAIA, BrowseComp, and xbench-ds) to demonstrate ScalingInter-RL's applicability in continuous high-dimensional spaces or real-world environments
> >
> > We hope that these additions have fully addressed your questions. If so, we would be grateful if you would consider updating your final rating. We sincerely appreciate your time and further feedback.
> >
> > Best regards,
> >
> > Authors of ICLR 2026 Submission 20092

---

### Official Review · Reviewer_M63s · 2025-10-31

**Soundness:** 4
**Presentation:** 3
**Contribution:** 3
**Rating:** 10
**Confidence:** 4

**Summary:**

This work introduces a new framework for multi-turn LLM RL and a multi-staged training approach for long-horizon RL by progressively scaling interaction lengths. The results are strong relative to baselines, often outperforming much larger models.

**Strengths:**

- This work tackles challenging LLM RL tasks with real environment interactions, which is underexplored relative to the more common math reasoning. The proposed framework would be very useful for the LLM RL community.
- ScalingInter-RL is well-motivated and a natural solution to the training instability of long-horizon RL
- The results are impressive and demonstrate the strength of the proposed method/framework over strong baselines.

**Weaknesses:**

- The text in some of the figures is small relative to the main text (Figure 1, 5, 6, 7)

**Questions:**

- Were any alternatives considered to the current linear schedule for the maximum number of interactions at each training step? In particular, I'd be curious to hear if this can be dynamically adapted (for instance allowing different RL rollouts to have slightly different limits and seeing which ones perform best).

---

> ### Author Response · Authors · 2025-11-20
> **Response to Reviewer M63s.**
>
> Thank you very much for your valuable suggestions and your strong recognition of our work!
>
> 1. **Question about using different schemes to adjust the maximum number of interactions in ScalingInter-RL.**
>    - In  **Point #2 of the General Response** , we adopted different schemes to dynamically adjust the maximum number of interactions in ScalingInter-RL. We found that fine-grained, dynamic adjustment can indeed bring slight performance improvements, which provides us with new insights; we plan to further explore this direction in future work.
> 2. **Question about the font size in the figures and tables.**
>    - Thank you very much for this suggestion. Due to space limitations, some of the figures are indeed displayed relatively small, and we have also moved part of the experiments to the appendix. In subsequent versions of the manuscript, we will follow your advice to adjust the layout, enlarge the figures, and make them clearer.
>
> Once again, we sincerely appreciate your recognition and feedback — it means a great deal to us. We will do our best to improve the manuscript according to your suggestions.

---

### Author Response · Authors · 2025-11-20
**General Response. Part [3/3]**

### #6 Experiments of applying ScalingInter-RL to other frameworks.

Following Reviewer yP1T’s suggestion, we conducted experiments in another framework. Specifically, we adopted the RAGEN framework [10] and implemented the BabyAI task. The experimental results are shown in the table below. **We find that ScalingInter-RL performs well across different frameworks.**

| Method                | Performance |
| --------------------- | ----------- |
| Base Model            | 69.7        |
| RAGEN-RL              | 82.5        |
| RAGEN+ScalingInter-RL | 88.4        |

### #7 Analysis of HTTP interaction overhead.

Following the suggestion of Reviewer gQCC, we analyzed the HTTP network communication overhead in our framework, i.e., the difference between deploying the environment and the model on different machines versus on the same machine, and we measured the proportion of communication time per round relative to the total rollout time. **As shown in the table below, the ratio of HTTP communication time to rollout time is extremely small, indicating that the time cost of HTTP communication does not affect the overall training efficiency.** Moreover, considering the benefits of HTTP communication (flexibility, scalability, etc.), we regard this as a favorable design choice.

| Environment | HTTP Communication Time | Total Rollout Time | Percentage |
| ----------- | ----------------------- | ------------------ | ---------- |
| TextCraft   | 0.00436 s               | 0.355 s            | 1.22%      |
| BabyAI      | 0.00191 s               | 0.203 s            | 0.94%      |
| SciWorld    | 0.00278 s               | 0.192 s            | 1.44%      |

### #8 Analysis of computational resources and efficiency.

Following the suggestions of Reviewers gQCC and yP1T, we analyzed the efficiency of ScalingInter-RL by examining the training reward, the cumulative number of interaction rounds during training, and the total training time. As shown in  **the Figure 10 in the revised manuscript** , we can observe that, thanks to the stage-based design, **ScalingInter-RL is able to achieve relatively high rewards with comparatively high efficiency and reduced time and resource consumption.**

[1] REINFORCE++: Stabilizing Critic-Free Policy Optimization with Global Normalization

[2] DeepSeekMath: Pushing the Limits of Mathematical Reasoning in Open Language Models

[3] https://miromind.ai/blog/miromind-research-agent

[4] WebShaper: Agentically Data Synthesizing via Information-Seeking Formalization

[5] Cognitive Kernel-Pro: A Framework for Deep Research Agents and Agent Foundation Models Training

[6] DeepResearcher: Scaling Deep Research via Reinforcement Learning in Real-world Environments

[7] GAIA: A BENCHMARK FOR GENERAL AI ASSISTANTS

[8] BrowseComp: A Simple Yet Challenging Benchmark for Browsing Agents

[9] https://xbench.org/

[10] RAGEN: Understanding Self-Evolution in LLM Agents via Multi-Turn Reinforcement Learning

---

### Author Response · Authors · 2025-11-20
**General Response. Part [2/3]**

### #3 Experiments of the design for initial-stage of ScalingInter-RL.

Following the suggestion of Reviewer odrg, we adopted several additional approaches in the initial stage to investigate, on the Deep Search task, whether these new methods based on subtask design or reward function design can bring performance gains:

* **New strategy 1:** We first classify queries via AVG@16 sampling into easy queries (accuracy 70%–100%), medium queries (30%–70%), and hard queries (below 30%), and construct a difficulty-based curriculum of subtasks, i.e., we train on easy queries in the early phase and on hard queries in the later phase.
* **New strategy 2:** We randomly shuffle all the data and assign correctness rewards of 0.6 for easy data, 0.8 for medium data, and 1.0 for hard data.

The experimental results are as follows:

| Model           | Performance |
| --------------- | ----------- |
| Base Model      | 18.8        |
| AgentGym-RL-7B  | 34.0        |
| ScalingInter-7B | 38.3        |
| New strategy 1  | 38.6        |
| New strategy 2  | 37.8        |

We find that carefully designing these components can yield modest performance gains (i.e., 0.3). However, this approach requires annotating the training data by difficulty before training, which introduces additional resource overhead. The scheme we adopt in the paper is much simpler and, even without such fine-grained tuning, already surpasses the basic RL baseline. This reveals a trade-off between design complexity and performance.

### #4 Experiments of applying ScalingInter-RL to more algorithms.

Following the reviewers’ suggestions, we applied ScalingInter-RL to other algorithms and present the results below. We can observe performance differences across algorithms, which is consistent with previous work [1][2]. **In addition, it is clear that ScalingInter-RL brings performance improvements across different algorithms.**

| RL Algorithm    | Method         | TextCraft | BabyAI | SciWorld |
| --------------- | -------------- | --------- | ------ | -------- |
| Base Model      | -              | 42.00     | 66.67  | 1.50     |
| PPO             | AgentGym-RL-7B | 68.00     | 86.66  | 10.83    |
|                 |ScalingInter-7B | 71.00     | 90.00  | 25.69    |
| REINFORCE++     | AgentGym-RL-7B | 73.00     | 84.44  | 13.63    |
|                 |ScalingInter-7B | 77.00     | 87.77  | 26.14    |

### #5 Experiments of generalizing ScalingInter-RL to more real-world tasks.

In the paper, we have conducted experiments on realistic WebArena and SearchQA tasks and achieved reasonably strong results. Here, we go one step further by using data from [3][4][5] and a Google search engine based on the Serper API [6] to evaluate performance on GAIA-Text-103 [7], BrowseComp [8], and xbench-ds [9] (using the AVG@3 metric). **As shown in the table below, ScalingInter-RL yields additional performance gains.**

|                 | GAIA_Text-103 | Browsecomp | xbench-ds | Average |
| --------------- | ------------- | ---------- | --------- | ------- |
| AgentGym-RL_10  | 36.9          | 3.3        | 21.7      | 20.6    |
| AgentGym-RL_20  | 38.5          | 6.5        | 27.1      | 24.0    |
| AgentGym-RL_30  | 37.6          | 7.7        | 29.5      | 24.9    |
| ScalingInter-RL | 43.3          | 8.5        | 31.5      | 27.8    |

---

### Author Response · Authors · 2025-11-20
**General Response. Part [1/3]**

Dear reviewers, many thanks for your time in reviewing our work. We are truly encouraged by your positive evaluation of the value and contribution of our paper. We sincerely appreciate your recognition of the clarity of our motivation, the generality of our framework, and the solidness of our experiments. We hope our work can provide insights and contribution to the community.

We revised the manuscript accordingly, adding several experiments, analyses, tables, and figures. The tables are all included in the response of rebuttal, while the figures are presented in the manuscript.

In what follows, we provide a general response addressing several issues of common interest to multiple reviewers, and we also include additional experiments in the hope of resolving your concerns.

### #1 Experiments of ablation study for ScalingInter-RL.

Following the reviewers’ suggestions, we conducted detailed ablation studies on the initial number of interaction rounds, the stage transition frequency, and the interaction interval. We carried out these experiments on Deep Search, and the results are as follows:

| Interact Turn List | Stage Transition Frequency | Performance |
| ------------------ | -------------------------- | ----------- |
| [5,8,10]           | 100                        | 38.3        |
| [5,8,10]           | 75                         | 37.8        |
| [5,8,10]           | 125                        | 38.5        |
| [3,8,13]           | 100                        | 36.8        |
| [8,10,12]          | 100                        | 37.6        |
| [5,10,15]          | 100                        | 39.1        |
| [5, 7, 9]          | 100                        | 37.8        |

### #2 Experiments of using different schemes to adjust the maximum number of interactions in ScalingInter-RL.

Following the suggestion of Reviewer M63s, we adopted several alternative schemes for this adjustment.

* Scheme 1: We select a validation set and validate every 50 steps. If the accuracy exceeds a certain threshold or the model has been trained for a duration of more than 150 steps, we increase the Max Turn Number; otherwise, we keep the Max Turn Number unchanged.
  | Steps            | 0-150 | 150-250 | 250-500 |
  | ---------------- | ----- | ------- | ------- |
  | Max Turn Number  | 5     | 8       | 10      |
  | Evaluation Score | 33.3  | 37.5    | 39.4    |
* Scheme 2: Similar to an oscillatory function, we dynamically increase and decrease the Max Turn Number every 50 steps.
  | Steps            | 0-50 | 50-100 | 100-150 | 150-200 | 200-250 | 250-300 | 300-350 | 350-400 | 400-450 | 450-500 |
  | ---------------- | ---- | ------ | ------- | ------- | ------- | ------- | ------- | ------- | ------- | ------- |
  | Max Turn Number  | 5    | 8      | 10      | 8       | 5       | 8       | 10      | 8       | 5       | 8       |
  | Evaluation Score | 30.8 | 33.5   | 37.9    | 37.1    | 35.3    | 36.0    | 37.8    | 35.5    | 30.5    | 32.5    |

The experimental results show that Scheme 1 can better monitor the training dynamics, with switches occurring at steps 150 and 250, ultimately achieving a performance of 39.4, which is somewhat higher than our initial result. Scheme 2, however, is relatively more oscillatory, and its best performance does not surpass that of our relatively fixed switching strategy. Note that in the paper, since it already surpasses the RL baselines, we initially adopted the basic scheme as the default.

---

### Author Response · Authors · 2025-12-02
**Summary**

We sincerely thank the reviewers, the Area Chair, the Senior Area Chairs and Program Chairs for the valuable feedback. We also deeply value thoughtful evaluations, which highlight several strengths of our works:

1. Clear and meaningful problem motivation, addressing the lack of unified RL frameworks for training LLM agents in diverse, real-world multi-turn environments (Reviewer M63s, odrg, gQCC, yP1T)
2. Modular, flexible, and end-to-end AgentGym-RL framework that lowers the barrier for agent research, supports multiple algorithms, and offers strong engineering design for reproducibility and extensibility (Reviewer odrg, gQCC, yP1T)
3. Well-motivated and intuitive ScalingInter-RL method that stabilizes long-horizon training by progressively expanding interaction horizons, effectively balancing exploration depth and training stability (Reviewer M63s, odrg, gQCC)
4. Strong and comprehensive evaluation results across 5 environments and 27 tasks, showing substantial gains over strong baselines and even outperforming or matching much larger proprietary models, providing robust evidence of generality and effectiveness (Reviewer M63s, odrg, gQCC, yP1T)

A major concern shared by reviewers is the hyperparameter sensitivity of the ScalingInter-RL method. Another common concern relates to its generalizability, including how well it transfers to other RL algorithms, more real-world tasks and alternative frameworks. Reviewers also noted the need for analysis of the computational cost of ScalingInter-RL and HTTP interaction overhead of AgentGym-RL framework.

In our rebuttal, we conduct additional experiments and offer clear analysis to address these concerns, including:

1. Comprehensive ablations on ScalingInter-RL, including the initial interaction rounds, stage transition frequency, interaction interval, horizon-adjustment schemes, and the initial-stage design based on subtask or reward function.
2. Generalization of ScalingInter-RL to PPO, REINFORCE++, the RAGEN framework, and diverse real-world environments including GAIA-Text-103, BrowseComp, and xbench-ds.
3. Comparison of ScalingInter-RL with the (curiosity-based) NGU exploration technique from classical RL.
4. Case study comparing short-horizon agents, long-horizon agents, and ScalingInter-RL, along with an empirical analysis of the instability in long-horizon training.
5. Measurement of communication overhead by quantifying the proportion of HTTP interaction time per round relative to total rollout time. The results show negligible communication overhead during interaction.
6. Analysis of training efficiency through training reward dynamics, cumulative interaction rounds, and end-to-end training duration. The results demonstrate the high efficiency of ScalingInter-RL.

All these results are consistently encouraging, providing further validation of the effectiveness and generalizability of ScalingInter-RL, while also demonstrating the efficiency of the AgentGym-RL framework.

**Our rebuttal received positive recognition from Reviewer yP1T and gQCC. Both of them  acknowledged that we have successfully addressed  their questions and concerns.** During the discussion phase, Reviewer yP1T raised the overall score to 8, and Reviewer gQCC increased the contribution score. Although other reviewers did not participate in the discussion, we made every effort to address their concerns through detailed responses and additional experiments.

We have revised our manuscript based on the rebuttal content to further strengthen our work. Finally, we greatly appreciate the reviewers' recognition of AgentGym-RL and ScalingInter-RL and their constructive suggestions. We hope that our work can provide insights and contribute to the broader development of agent research.

---

### Meta-Review · Area_Chair_HyAn · 2026-01-01

**Summary:**

This paper presents a well-designed open-source framework for training LLM agents via multi-turn RL, together with a staged strategy for stabilising long-horizon training. Reviewers raised concerns about hyperparameter sensitivity, generality across algorithms and environments, and computational overhead, all of which were convincingly addressed in the rebuttal through extensive ablations, cross-algorithm and cross-framework validation, efficiency analysis, and overhead measurements. Multiple reviewers explicitly raised their scores after discussion. Overall, this is a solid and timely contribution with clear practical impact for the agent-RL community.

**Reviewer Concerns:**

Concerns about hyperparameter sensitivity, generalisation across algorithms/environments, training stability, and computational/communication overhead were addressed through additional ablations, cross-framework experiments, and efficiency analyses. However, there are still limitations regarding theoretical justification and the largely engineering-focused nature of the contribution, but these do not affect the empirical validity.

**Reviewer Scores:**

Overall, the reviewers' comments with positive scores and the authors rebutall would converge toward a consensus acceptance.

---

### Decision · Program_Chairs · 2026-01-26

Accept (Oral)